# CPT2 Deficiency Modeled in Zebrafish: Abnormal Neural Development, Electrical Activity, Behavior, and Schizophrenia-Related Gene Expression

**DOI:** 10.3390/biom14080914

**Published:** 2024-07-26

**Authors:** Carly E. Baker, Aaron G. Marta, Nathan D. Zimmerman, Zeljka Korade, Nicholas W. Mathy, Delaney Wilton, Timothy Simeone, Andrew Kochvar, Kenneth L. Kramer, Holly A. F. Stessman, Annemarie Shibata

**Affiliations:** 1Department of Biomedical Sciences, Creighton University, Omaha, NE 68178, USA; carly.elaine.baker@gmail.com (C.E.B.); kenkramer@creighton.edu (K.L.K.); 2Department of Biology, Creighton University, Omaha, NE 68178, USA; aaronmarta@creighton.edu (A.G.M.); nathanzimmerman@creighton.edu (N.D.Z.); nicholasmathy@creighton.edu (N.W.M.); delaneyw7@gmail.com (D.W.); andrewkochvar@creighton.edu (A.K.); 3Department of Pediatrics, Department of Biochemistry & Molecular Biology, College of Medicine, University of Nebraska Medical Center, Omaha, NE 68178, USA; zeljka.korade@unmc.edu; 4Department of Pharmacology and Neuroscience, Creighton University, Omaha, NE 68178, USA; timothysimeone@creighton.edu (T.S.); hollystessman@creighton.edu (H.A.F.S.)

**Keywords:** carnitine palmitoyltransferase, CPT2 deficiency, zebrafish, β-oxidation, brain development, seizure-like activity, schizophrenia-related gene expression, neurodegenerative disease

## Abstract

Carnitine palmitoyltransferase 2 (CPT2) is an inner mitochondrial membrane protein of the carnitine shuttle and is involved in the beta-oxidation of long chain fatty acids. Beta-oxidation provides an alternative pathway of energy production during early development and starvation. CPT2 deficiency is a genetic disorder that we recently showed can be associated with schizophrenia. We hypothesize that CPT2 deficiency during early brain development causes transcriptional, structural, and functional abnormalities that may contribute to a CNS environment that is susceptible to the emergence of schizophrenia. To investigate the effect of CPT2 deficiency on early vertebrate development and brain function, CPT2 was knocked down in a zebrafish model system. CPT2 knockdown resulted in abnormal lipid utilization and deposition, reduction in body size, and abnormal brain development. Axonal projections, neurotransmitter synthesis, electrical hyperactivity, and swimming behavior were disrupted in CPT2 knockdown zebrafish. RT-qPCR analyses showed significant increases in the expression of schizophrenia-associated genes in CPT2 knockdown compared to control zebrafish. Taken together, these data demonstrate that zebrafish are a useful model for studying the importance of beta-oxidation for early vertebrate development and brain function. This study also presents novel findings linking CPT2 deficiency to the regulation of schizophrenia and neurodegenerative disease-associated genes.

## 1. Introduction

In total, inborn errors in metabolism, or congenital metabolic diseases, affect approximately 1 in every 800 live births [1,2]. Congenital metabolic diseases are inherited genetic disorders that primarily affect enzymes needed to convert food into energy. Fatty acid disorders, such as CPT2 deficiency, that disrupt lipid utilization are one type of metabolic disease [3,4,5]. Metabolic diseases and fatty acid disorders can be associated with neurological dysfunction such as epilepsy, attention deficit disorders, intellectual disabilities, and autism [1,2].

The βeta (β)-oxidation of long-chain fatty acids (LCFAs) is dependent upon the carnitine palmitoyl transferase (CPT) system. The CPT system includes the enzymes carnitine palmitoyltransferase 1 (CPT1) that catalyzes the transfer of acyl groups from acyl-CoA to carnitine in the outer mitochondrial membrane and carnitine palmitoyltransferase 2 (CPT2) that converts the long-chain acylcarnitine to acyl-CoA for oxidation in the inner mitochondrial membrane. During early brain development and periods of starvation, neurons rely on fatty acid β-oxidation for oxidative bioenergetic metabolism [6,7]. CPT1 and CPT2 are highly expressed in neurons and astrocytes of the hypothalamus, amygdala, hippocampus, brainstem, and spinal cord [8,9,10,11]. The synthesis of acyl-CoA by CPT2 is important for cellular signaling, antioxidant activity, transcriptional regulation, and cholinergic neurotransmission [9]. 

Commonly, CPT2 deficiency is characterized as one of three distinct clinical presentations: lethal neonatal, severe infantile hepatocardiomuscular, and myopathic forms. Lethal neonatal and severe infantile forms are classified by multisystemic dysfunction involving liver failure, low ketone levels, cardiomyopathy, seizures, and early death. The myopathic form is most common and manifests throughout life. People with myopathic CPT2 deficiency are known to experience fatigue, muscle pain, and weakness after periods of exercise or starvation [12,13,14]. Our work described a proband with confirmed CPT2 deficiency who presented with mild-to-moderate symptoms that included seizure during early childhood, neurocognitive deficits in adolescence, and schizophrenia as a young adult [15]. While CPT1 deficiency is associated with neurocognitive deficits, CPT2 deficiency had not been associated with neurocognitive deficits and schizophrenia [15,16]. Recently, high levels of acylcarnitine in the plasma of schizophrenic individuals has been reported [17,18]. CPT2 mutation may underlie our proband’s epilepsy, neurocognitive challenges, and later-presenting schizophrenia. The biological mechanisms by which CPT2 mutations that cause dysfunction of the carnitine shuttle and disrupted β-oxidation contribute to abnormal brain development and function are not fully understood. Thus, there is a critical need to establish model systems that dissect the role of CPT2 mutation in early neurodevelopment.

The zebrafish (Danio rerio) represents an ideal vertebrate model system for studying human metabolic disease [19,20]. Approximately 70% of human genes have at least one zebrafish orthologue [21]. Human *Cpt2* (ENSG00000157184) and zebrafish *cpt2* (ENSDARG00000038618) are homologous (www.zfin.org) and the sequences show 70.9% alignment as determined by Expasy (www.expasy.org). Zebrafish body and brain development is rapid and well-studied. Zebrafish exhibit highly conserved physiological processes and defined behavioral traits. The maintenance of developing zebrafish is relatively inexpensive [22]. Gene silencing techniques, such as morpholino knockdown, are effective in zebrafish and can be readily used to examine protein function. These characteristics of zebrafish also make it a powerful high-throughput system for evaluating druggable targets for disease. We and others propose that zebrafish can be used to investigate metabolic disease, such as CPT2 deficiency, and the association with neurodevelopmental and neurodegenerative disorders in humans [22,23,24]. 

We hypothesized that mild-to-moderate CPT2 deficiency disrupts lipid and acylcarnitine signaling early in development, affecting gene transcription and causing abnormal neural network formation, electrical activity, and behavior. Translation blocker (TB) and splice blocker (SB) morpholino oligonucleotides (MO) generated two zebrafish knockdown models with mild or moderate reductions in CPT2 expression. These *cpt2* knockdowns allowed for comprehensive analyses and confirmation that CPT2 deficiency can be effectively modeled using the vertebrate zebrafish system. The zebrafish system revealed that the inability to properly utilize fatty acids and acylcarnitine for β-oxidation during early development leads to abnormal brain structure and function and the upregulation of genes homologous to those associated with schizophrenia and other neurodevelopmental and neurodegenerative diseases in humans.

## 2. Materials and Methods

All reagents and their sources are in Appendix B. 

### 2.1. Zebrafish Husbandry

Adult Tu/AB zebrafish were housed in Creighton University’s Animal Research Facility and bred to produce embryos for microinjections following animal protocol #0924. Post-breeding, fertilized embryos were injected at the single-cell stage and larvae were raised to 5 days post injection (dpi) in E3 media (1X autoclaved solution of 5 mM NaCl, 0.17 mM KCl, 0.33 CaCl_2_, 0.33 mM MgSO_4_, and 0.1% Methylene Blue) at 28 °C and 21.0% CO_2_. All procedures involving embryos and larvae followed animal protocol #1134. Animal protocols were approved by the Creighton University Institutional Animal Care and Use Committee.

### 2.2. Morpholino Injection of Zebrafish Embryos

Translation blocking (TB) and splice blocking (SB) knockdown morpholinos (MOs) for *cpt2* and a standard scrambled control MO (CTRL, PCO-StandardControl-100-F) were generated and tagged with fluorescein by GeneTools, LLC (Philomath, OR, USA) as previously described (Appendix B, Table A2) [25]. Morpholino (MO) constructs were microinjected into zebrafish embryos at the single-cell stage using a modified protocol [26]. Briefly, morpholinos were diluted to a final concentration of 0.5, and 1.0 mM in near-isotonic solution 958 mM NaCl, 0.7 mM KCl, 0.4 mM MgSO_4_, 0.6 mM Ca(NO_3_)2, and 5.0 mM HEPES, pH 7.6 [25]. MOs were heated at 65 °C for 10 min and centrifuged for 2 min at 12,000 RPM. Injection needles were fabricated by pulling aluminosilicate glass capillaries on a Flaming/Brown Micropipette puller model, P-97 (Sutter Instruments, Novato, CA, USA). Needles were filled with 4 µL of the MOs, clipped for injection, and mounted onto a micromanipulator (Sutter Instruments, Novato, CA, USA). Embryos at the single-cell stage were aligned on a 1.5% agar plate and visualized on a Leica M165C microscope for injection. A nitrogen-based Narishige injector (Narishige (International) Ltd., London, UK) with a Harvard apparatus (PLI-90) (Harvard Apparatus, Holliston, MA, USA) was used to inject approximately 1.5 nL of MOs into each embryo. After injection, embryos were gently washed out of the injection plate using an E3 medium and grown for 1 dpi at 28 °C.

### 2.3. Assessment of Morpholino-Injected Zebrafish

At 1 dpi, injected embryos were examined for uptake of fluorescently tagged MOs using a Zeiss NeoLumat S 1.5x FWD 30 mm (ZEISS Microscopy Customer Center, Dublin, CA, USA). Non-fluorescent embryos were euthanized following protocol #1134 and were not used for any experiments. Each group of 0.5 and 1.0 mM-injected and fluorescent embryos were examined for viability and developmental progression. Opaque, dense, coagulated zebrafish were counted as dead and were used to determine percent viability of each animal group. 

### 2.4. RNA Isolation and cDNA Synthesis

To collect RNA, larvae at 5 dpi were euthanized on ice and placed in microcentrifuge tubes where the excess E3 media were removed. Ceramic beads (BioSpec, Bartlesville, OK, USA) and TRI Reagent™ Solution (Invitrogen, Waltham, MA, USA) were added to larvae to fill a 1.5 mL microcentrifuge tube. Tubes were homogenized in a high-energy cell disrupter BioSpec Mini-Beadbeater-96 three times for 30 s at 4 °C. After homogenization, beads were removed, and samples were centrifuged for 15 min at 13,200 RPM at 4 °C. The supernatant was removed, and chloroform was added to the homogenate at a 1:3 ratio. Samples were vortexed for 15 s, incubated at RT for 5 min, and centrifuged for 30 min at 13,200 RPM at 4 °C. The aqueous layer was collected and mixed with equal parts of isopropanol. Samples were vortexed for 30 s and incubated at room temp for 15 min and then centrifuged for 20 min at 13,200 RPM at 4 °C. Precipitated RNA was resuspended in nuclease-free water. RNA concentration was determined using DeNovix DS-11 Spectrophotometer (DeNovix Inc., Wilmington, DE, USA) and RNA was stored at −80 °C. cDNA was generated using iScript (Bio-Rad, Hercules, CA, USA) following the Bio-Rad protocol. cDNA was diluted at a 1:5 ratio with nuclease-free water and stored at −20 °C.

### 2.5. RT-PCR

RT-PCR was used to determine the efficacy and specificity of the splice blocking (SB) *cpt2*-specific morpholino. Primer sets for *cpt2* and *actin* were purchased from Integrated DNA Technologies. The parameters for the PCR reaction were 98 °C for 10 s, 98 °C for 1 s, 50 °C for 5 s, and 72 °C for 1 min cycled through 39X followed by 72 °C for 1 min. The product of the PCR reaction was visualized on 1% TAE agarose gel using an SYBR Safe Gel stain (Thermo Fisher Scientific, Carlsbad, CA, USA). The gel was imaged with UV using a ChemiDoc XRS machine (Bio-Rad, Hercules, CA, USA).

### 2.6. Mitochondrial Isolation

Mitochondrial isolation was performed using 200 larvae per condition for each of 3 trials and mitochondrial protein was used with Western blotting as previously described [27]. Briefly, 5 dpi larvae were collected and euthanized on ice. Larvae were placed in a pre-chilled glass Dounce homogenizer with 1 mL of an ice-cold MB buffer (210 mM mannitol, 70 mM sucrose, 1 mM EDTA, 10 mM HEPES (pH 7.5)) plus a Halt protease inhibitor (Thermo Fisher Scientific, Carlsbad, CA, USA) and homogenized on ice using 10 passes in a Dounce homogenizer. The homogenate was incubated on ice for 10 min and then centrifuged at 300 G for 5 min. The supernatant was removed, and the pellet was resuspended in 1 mL of the ice-cold MB buffer. The pellet was disrupted on ice by 50 passes through a 26-gauge × 2/3 needle on a 1 mL syringe. The homogenate was centrifuged twice for 10 min at 1500 G to eliminate nuclei. The supernatant, containing mitochondria, cytosol, and microsomes, was centrifuged twice for 10 min at 10,600 G to pellet the mitochondria. The mitochondrial pellet was resuspended in 1 mL of the ice-cold MB buffer, centrifuged at 10,600 RPM for 10 min, resuspended in 30 µL of the MB buffer, and stored at −20 °C until use for Western blotting.

### 2.7. Larval Protein Extraction

Protein was extracted from 30 CTRL-, TB-, and SB-injected larvae at 5 dpi for each of 4 trials. Total protein extraction was performed as previously described [27]. Briefly, 30, 5 dpi, larvae were collected and euthanized on ice. Larvae were homogenized in a Pierce RIPA buffer (Thermo Fisher Scientific, Carlsbad, CA, USA) plus Halt protease inhibitors at a ratio of 1.75 µL of the buffer per fish. Samples were centrifuged at 4 °C for 10 min, and the supernatant was transferred to a new pre-chilled microcentrifuge tube and stored at −80 °C until use for Western blotting.

### 2.8. Western Blot

Following mitochondrial isolation or larval protein extraction, total protein was measured using Pierce™ BCA Protein Assay Kit (Thermo Fisher Scientific, Carlsbad, CA, USA). Between 100 and 200 µg of protein was denatured in a 2X (Bio-Rad, Hercules, CA) or 4X (Thermo Fisher Scientific, Carlsbad, CA, USA) sample buffer with 5% DTT (Thermo Fisher Scientific, Carlsbad, CA, USA). Protein samples were vortexed and heated at 95 °C for 5 min. Protein samples were loaded onto 4–20% SDS-PAGE gradient Mini TGX gels (Bio-Rad, Hercules, CA, USA) and separated by electrophoresis (Bio-Rad, Hercules, CA, USA; Mini-PROTEAN Tetra System). Precision Plus Western C Blotting Standard (Bio-Rad; Hercules, CA, USA) was used. In total, 10 µg of human CPT2 protein (Abnova, Walnut, CA) was used as a positive control. Gels were run in 1X TGS Running Buffer (Thermo Fisher Scientific, Carlsbad, CA, USA) and transferred onto Trans-Blot Turbo PVDF Transfer Pack membranes (Bio-Rad, Hercules, CA, USA) in a Bio-Rad Trans-Blot Turbo system. After transfer, membranes were washed 3 times in 1X TBST for 5 min and blocked in EveryBlot Blocking Buffer (Bio-Rad, Hercules, CA, USA) for 1 h at RT. Membranes were incubated with primary antibodies overnight at 4 °C. Primary antibodies used were CPT2 (Bioss, Woburn, MA, USA), phosphorylated α-synuclein (Invitrogen, Waltham, MA, USA), α-synuclein (Invitrogen, Waltham, MA, USA), and actin (Invitrogen, Waltham, MA, USA) and were diluted at 1:1000. Membranes were washed in 1X TBST 3 times for 5 min. Membranes were incubated with an anti-rabbit secondary antibody (Invitrogen, Waltham, MA, USA) at a 1:6000 dilution in a blocking buffer and HRP conjugate to detect Precision Plus Protein Standard (Bio-Rad, Hercules, CA, USA) at a 1:3000 dilution for 90 min. Membranes were washed 3 times in 1X TBST for 5 min. Super Signal West Pico PLUS Chemiluminescent was used for detection. Membranes were imaged and analyzed using a ChemiDoc XRS machine and software (Bio-Rad, Hercules, CA, USA).

### 2.9. Liquid Chromatography in Tandem with Mass Spectrometry

LC-MS/MS was performed using 30 zebrafish at 5 dpi per condition for 5 experiments. To determine the acylcarnitine profile of zebrafish, 5 dpi larvae were euthanized on ice and transferred to a 1.5 mL microcentrifuge tube. E3 media were removed, and larvae were frozen in liquid nitrogen. The acylcarnitine profile was assessed following previously published protocols [28]. In brief, samples were spiked with a known amount of d3-palmitoylcarnitine as the internal standard and lipids were extracted. After extraction, acylcarnitines were injected into the column (Phenomenex Luna Omega C18, 1.6 µm, 100 A, 2.1 mm × 100 mm) with 100% acetonitrile (0.1% *v*/*v* formic acid) (solvent B) and water/acetonitrile (90:10 *v*/*v*, 0.1% *v*/*v* formic acid, and 10 mM ammonium formate) (solvent A) for a 5 min runtime at a flow rate of 500 µL/min. Acylcarnitine species were analyzed by SRM using transitions of the precursor ion (as M + H) to the respective product ions with an 85 *m*/*z* ratio. The analysis was performed with TraceFinder software version 4.1 (Princeton, NJ, USA).

### 2.10. Morphological Evaluation

Morphological evaluation was performed on 25, 5 dpi, larvae per condition for 2 trials. Zebrafish were collected, euthanized on ice, and fixed in 4% paraformaldehyde (PFA) in phosphate-buffered saline (PBS) overnight at 4 °C. Larvae were analyzed through quantitative and qualitative analyses. Qualitative analyses of morphological characteristics included the presence of abnormal yolk sac shape, pericardial edema, and curved tail. An abnormal yolk sac was characterized by swelling around the abdomen. Pericardial edema was characterized by swelling of the cardiac region located anterior to the yolk sac. A curved tail was characterized by an abnormal curvature of the tail. Quantitative measurements included the determination of eye length, eye distance, rump length, and standard length. Eye length spanned from the anterior portion of the eye to the posterior portion. Rump length spanned from the anterior portion of the head to the anus. Standard length spanned from the most anterior portion of the head to the most posterior portion of the tail. Eye distance was recorded dorsally as the distance along the most lateral position from one eye to the other eye. Images were acquired using Olympus Life Science EPview Image Analysis Software Version: 510 (Center Valley, PA, USA). The fish were examined under Olympus SZX16 Model Number: SZX2-ILLT using a 3.2Xobjective. Images were acquired with an Olympus EP50 camera. For the image analysis, the poly-line function within Olympus Life Science CellSens Dimension Life Science Imaging Software Version 1.18 was used (Center Valley, PA, USA).

### 2.11. Oil Red O Whole-Mount Stain

Oil Red O staining of lipids was performed using 10 larvae per condition. The whole-mount Oil Red O stain was modified as previously described [29]. Briefly, 5 dpi larvae were euthanized on ice and fixed in 4% PFA in PBS in a 12-well plate at 4 °C overnight. PFA was removed and larvae were washed twice with 1X PBS for 5 min. After washing, a 1.0 mL 0.5% oil red solution in 60% isopropanol was added for no more than 15 min. The Oil Red O solution was removed and the larvae were rinsed with Epure twice for 5 min. Larvae were stored in Epure at RT. Larvae were mounted in 70% glycerol and imaged with a 6X objective on Olympus SZX16 Model Number: SZX2-ILLT (Olympus Corporation, Tokyo, Japan) equipped with an Olympus EP50 camera. Mean color intensity as determined by the formula 100—sample pixel intensity/background pixel intensity) was measured using greyscale images and the quantification of pixel intensity on Fiji ImageJ Version: 2.0.1 (Bethesda, MD, USA).

### 2.12. Alcian Blue Whole-Mount Stain and Cartilage Measurement

Alcian blue staining was performed as previously described [30]. Briefly, 5 dpi larvae for each condition were euthanized on ice and fixed in 4% PFA in PBS at 4 °C overnight. Fixed zebrafish were washed twice with phosphate-buffered saline with 0.1% Tween (PBST) for 10 min and incubated in 1.0 mL of 0.1% Alcian blue dye in 70% ethanol and 30% glacial acetic acid overnight at RT. Larvae were washed twice with 200-proof ethanol for 2 min and put through an ethanol series of 20 min wash steps of 180 Proof, 160 Proof, and 140 Proof ethanol. Larvae were rinsed in PBST for 20 min and then bleached in a 3% H_2_O_2_/1% KOH solution for 90 min. Larvae were incubated in 0.1% trypsin in 30% saturated aqueous sodium borate for 40 min. Bleaching and trypsin treatments were repeated twice. Larvae were rinsed 3 times in PBST for 5 min and stored in PBST for imaging. Larvae were imaged in 70% glycerol on Olympus SZX16 Model Number: SZX2-ILLT equipped with an Olympus EP50 camera. Cartilage formation in the zebrafish head was evaluated by measuring the distance between the Mechel’s and ceratohyal cartilage and the angles of both the Mechel’s and ceratohyal cartilage.

### 2.13. Whole Brain Morphology and Immunofluorescence Labeling

Acetyl-tubulin and tyrosine hydroxylase staining was performed using 7–16 brains per condition. Whole brain immunofluorescence (IF) was performed using whole mount IF labeling [31]. Briefly, euthanized, 5 dpi zebrafish larvae were fixed in Anatech Ltd. Prefer Fixative (Thermo Fisher Scientific, Carlsbad, CA, USA) for 48 hrs. Brains were rinsed in PBS 0.1% Tween-20, post-fixed in 4% PFA for 10 min, and permeabilized in 0.5% Triton X-100 in PBS. Larvae brains were bleached in 1.5% H_2_O_2_/1% KOH; blocked in 10% goat serum, 1% DMSO, and 1% BSA; and then incubated in a primary antibody (1:250 dilution) overnight at 4 °C. A secondary antibody was incubated (1:300 dilution) at 4 °C overnight. After rinsing in PBS+ 0.1% Tween 20, 4′,6-diamidino-2-phenylindole (DAPI) (Thermo Fisher Scientific, Carlsbad, CA, USA) staining was performed (1:5000 dilution) and brains were post-fixed in 4% PFA for 20 min, cleared, and stored in 90% glycerol in PBS. Brains were imaged on a Nikon Live Cell Eclipse TI-FL confocal microscope (Nikon Instruments Inc., Tokyo, Japan) in 30% glycerol. Images are presented as maximum-intensity z-projections generated by Nikon NIS-Elements software Version: 5.21 (Nikon Instruments Inc., Tokyo, Japan). The 355-laser power was 90% for DAPI; 488 nm laser power was 40% for tyrosine hydroxylase; and 488 nm laser power was 10% for acetyl-tubulin. LUT was at the max range for all images, 0 to 65,000. Primary antibodies used were anti-tyrosine hydroxylase (Millipore Sigma, Burlington, MA, USA) and anti-acetylated tubulin (Millipore Sigma, Burlington, MA, USA). Secondary antibodies used were anti-Rabbit-Alexa Fluor 488 (Abcam, Waltham, MA, USA), and anti-Mouse-Alexa Fluor 488 (Thermo Fischer Scientific, Carlsbad, CA, USA). The brain, brain regions, and cells were analyzed using NIS-Elements.

### 2.14. Viewpoint ZebraBox Behavioral Analysis

Behavioral assays were performed using a previously described light stimulation method [32]. Briefly, individual 5 dpi zebrafish were placed in wells of a 96-well plate. Plates were placed in the ZebraBox system (ViewPoint, Montreal, QC, Canada) and maintained at 28 °C. Zebrafish movement was stimulated using a series of light and dark periods over 40 min (first 10 min of light, followed by 20 min in dark, and then again 10 min of light). Light was at a target power of 100 or 0 for dark. Instantaneous edge transitions were used between the light-to-dark and dark-to-light phases. Two separate behavioral assays were performed using ViewPoint Software (Version 3.22). Quantization assays were performed with approximately 96 larvae per trial. Activity quantization was determined by quantifying the number of times larvae entered a particular activity state and how long they remained in that activity state. The activity states for this assay were freezing, medium movement, and burst movement. These activity states were monitored for the number of times the fish entered the states (count) and the time they spent in each activity state (duration). Thresholds for quantification were set such that sensitivity = 5, skip image count = 1, burst threshold = 275, and freezing threshold = 20. Tracking assays were performed with approximately 60 larvae per condition for two trials. Tracking assays monitored distance traveled to measure zebrafish inactivity, small movement, and large movement. Tracking movements were quantified by measuring the number of times fish entered each movement state (count), the duration they spent traveling in each movement state (duration), and the amount of distance they traveled in each movement state (distance). The thresholds set for analyses were set such that the low detection threshold = 30, skip image count = 1, large threshold = 8, inactivity threshold = 4, and jitter threshold = 0.

### 2.15. Electrophysiology Seizure Assay

Standard control and splice blocking MO larvae at 5 dpi were used (N = 3 per condition) for seizure assays that were modified from previously described work [33]. Briefly, larvae were anesthetized using an ice-cold E3 medium and were positioned so the dorsal side of the head was placed onto a MED64 microelectrode array comprised of 64 electrodes 20 × 20 µm in size and 100 µm apart (Alpha MED Scientific, Osaka, Japan). Larvae were secured with a tissue harp and fine nylon mesh (Warner Instruments, Holliston, MA, USA). Electrodes were 20 × 20 µm in size and 100 µm apart (MED64, Osaka, Japan). Larvae were acclimated to artificial cerebrospinal fluid (aCSF) (124 mM NaCl, 3 mM KCl, 26 mM NaHCO_3_, 2 mM CaCl_2_, 1 mM MgSO_4_, 1.25 mM NaH_2_PO_4_, 10 mM glucose, pH of 7.4) by perfusion for 10 min at RT. Recordings were acquired during perfusion of aCSF under constant flow at 1.8 mL/min of carbonation for 30 min. During the seizure assay, aCSF was supplied to larvae for the first 10 min of the recording, and for the next 10 min, 15 mM of PTZ, a GABA_A_ receptor antagonist (Sigma-Aldrich, St. Louis, MO, USA), in aCSF was added. PTZ is a noncompetitive antagonist of the gamma-aminobutyric acid (GABA_A_) receptor complex. PTZ is used to induce acute, severe seizures to examine seizure susceptibility in zebrafish. Glutamate is the major excitatory neurotransmitter and is released from activated neurons to bind NMDA, AMPA, and kainate receptors following release. For the final 10 min, 50 µM of AP5, an NMDA receptor antagonist (Sigma-Aldrich; St. Louis, MO, USA), and 50 µM of a CNQX and AMPA/kainate receptor antagonist (Sigma-Aldrich, St. Louis, MO, USA) in aCSF were perfused. Recordings were analyzed on Cambridge Electronic Design Spike 2 (Cambridge, UK). Electrode recordings were finite impulse response (FIR) bandpass filtered with the following parameters: low = 300 Hz, high = 3000 Hz, length = 1319. The root mean square (RMS) was determined. A single unit channel was created from the FIR-filtered channel by creating a new wave mark with the following parameters: left cursor = −0.4, right cursor = 0.9, top cursor = top of the page, bottom cursor = 4X RMS. The new channel was then converted to a mean squared wave mark trace of the 30 min recording. The spike rate was determined by averaging the spike amount per 1 s over each 10 min period.

### 2.16. RT-qPCR

RNA isolation and cDNA synthesis were performed before RT-qPCR. All primers were purchased from Integrated DNA Technologies (see Appendix B). Primers consisted of an equal mix of forward and reverse primers diluted with nuclease-free water at a stock concentration of 10 µM. A master mix was prepared with 10 µL of SYBR Green SuperMix (Bio-Rad, Hercules, CA, USA), 2 µL of a forward and reverse mixture of the primer of interest, and 3 µL of nuclease-free water per well. In total, 15 µL of the master mix and 5 µL of cDNA were used in each reaction. Plates were run in a Bio-Rad CFX PCR machine (Hercules, CA, USA) with the following parameters: 95 °C for 2 min, 95 °C for 5 s, 60 °C for 30 s, plate-read, and cycled 39X, and then 95 °C for 5 s, 65 °C for 31 s, 65 °C for 5 s + 0.5 °C/cycle and ramp at 0.5 °C/s, plate-read, and cycled 60X. Once completed, the plate was analyzed for gene expression changes using Bio-Rad CFX Maestro software version 4.1.2433.1219 (Hercules, CA, USA), and foldchange was determined using the 2^−∆∆Ct^ method.

### 2.17. Statistical Analysis

Ordinary one-way ANOVA was used for multiple comparisons for the analysis of CPT2 and acylcarnitine expression, morphological assessment, whole-mount staining quantification, behavior assays, and immunofluorescent quantification. An unpaired two-tailed *t*-test was performed for a single comparison. The *t*-test was used for RT-qPCR and electrophysiology data comparison. All errors between values were determined by the standard error of the mean (SEM). Differences were considered significant where *p* ≤ 0.05. All statistical analyses were performed using GraphPad Prism 9.1 software (SanDiego, CA, USA). 

## 3. Results

### 3.1. Development of a CPT2 Deficiency Model System in Zebrafish without Toxicity

To develop a vertebrate model system with mild-to-moderate CPT2 deficiency, Tu/AB *Danio rerio* zebrafish were injected with fluorescein-tagged MOs (Figure 1). SB MO was designed to bind to the pre-mRNA at the exon 5, intron 5 boundary, and the TB MO was designed to bind to the mature mRNA on exon 2 (Figure 1A). A scrambled CTRL MO was used as a control. Fertilized embryos at the single-cell stage were microinjected with 0.25, 0.5, and 1.0 mM CTRL, TB, or SB fluorescein-tagged MOs (Figure 1B). One day post injection (dpi), injected embryos were analyzed regarding fluorescein immunoreactivity, viability, and progression of development. The viability and progression of larval development following injection with 0.5 mM and 1.0 mM CTRL, TB, and SB MOs were compared to wildtype (WT) zebrafish. The WT zebrafish viability and progression of development were determined to be 77% at 1 day post fertilization. Zebrafish injected with 0.5 mM CTRL, TB, and SB MO showed 84%, 74%, and 79% viability and progression of development, respectively. Zebrafish injected with 1.0 mM CTRL, TB, and SB MO showed 60%, 59%, and 67% viability and progression of development, respectively. These results indicate that doubling the MO concentration resulted in a decrease in viability. The goal of this study was to evaluate a mild-to-moderate knockdown of *cpt2* translation. Since 0.5 mM MO-injected zebrafish were not less viable than WT and phenotypic effects were observed (see below), 0.5 mM MO-injected zebrafish were raised to 5 dpi (Figure 1C,D) for all subsequent experiments.

### 3.2. Evaluation of CPT2 Knockdown in Zebrafish Using Morpholinos

To evaluate CPT2 knockdown in TB and SB MO, RT-PCR and Western blot analyses were performed. The activity of SB MOs can be determined by RT-PCR because successful splice modification will result in a mobility shift or complete loss of the *cpt2* transcript. At 5 dpi, RNA was isolated from WT and MO-injected zebrafish. Forward and reverse primers for full-length *cpt2* were used (Appendix B, Table A3). The RT-PCR reaction showed that full-length *cpt2* (~1500 base pairs) cDNA is not amplified in SB-injected larvae while *cpt2* is amplified in uninjected WT and scrambled CTRL-injected larvae (Figure 2A). These data show a loss of the *cpt2* transcript and suggest that the SB MO inhibited pre-mRNA processing. Actin primers amplified actin cDNA in uninjected WT, standard CTRL-injected, and SB-injected zebrafish, supporting the specificity of SB MO targeting to *cpt2* pre-mRNA. Western blotting was used to confirm that TB and SB MO significantly reduced CPT2 protein expression in 5 dpi larvae. Mitochondria were isolated from 200 larvae per condition at 5 dpi, and 100 µg of protein from the mitochondrial fraction was used for Western blotting. A representative Western blot of CPT2 protein levels isolated from the mitochondrial fraction of 200 WT, CTRL-injected, TB-injected, and SB-injected zebrafish is shown (Figure 2B). Purified human CPT2 was used to verify specificity of the antibody for CPT2 (~74 kD). Zebrafish CPT2 migrates slightly higher at a molecular weight predicted to be closer to 80 kD (Figure 2B). The quantification of relative protein expression normalized to β-actin showed that scrambled CTRL MO did not reduce CPT2 expression in zebrafish compared to WT CPT2 levels (Figure 2C, *p* > 0.05). CPT2 expression decreased ~31% in TB-injected zebrafish compared to CTRL-injected zebrafish and ~38% compared to WT (N = 3, *p* > 0.05, Figure 2C). CPT2 expression significantly decreased by ~78% in SB-injected zebrafish compared to CTRL-injected zebrafish and ~85% compared to WT zebrafish (N = 3, *p* ≤ 0.05, Figure 2C). Western blot data confirm that MOs can generate a mild-to-moderate knockdown, and not a complete loss, of CPT2 protein in zebrafish. 

To confirm that knockdown of CPT2 affected the function of the carnitine shuttle system, liquid chromatography–mass spectrometry (LC-MS/MS) was used to analyze total acylcarnitine levels in WT, CTRL-injected, TB-injected, and SB-injected zebrafish (Figure 2D). Long-chain fatty-acylcarnitine species were not significantly different in CTRL-injected and WT zebrafish. Long-chain fatty-acylcarnitine species were significantly increased in TB-injected and SB-injected zebrafish compared to WT and CTRL-injected zebrafish. Total acylcarnitine levels increased ~70% (*p* <0.001) in TB-injected larvae and ~42% in SB-injected larvae compared to WT (*p* ≤ 0.05, Figure 2D). TB-injected acylcarnitine levels were significantly increased by ~55% compared to CTRL-injected zebrafish (*p* < 0.01, Figure 2D). SB-injected acylcarnitine levels increased ~28% compared to CTRL-injected zebrafish. This increase did not reach significance (*p* = 0.15, Figure 2D). The evaluation of the specific acylcarnitine species showed a significant change in expression for some acylcarnitine species compared to WT and CTRL-injected zebrafish (Appendix A). In particular, the data showed significantly increased levels of C16, C18, and C18:1. In particular, C16 levels in TB-injected zebrafish increased ~116% compared to WT (*p* < 0.0001, Appendix AC). C16 levels increased ~57% in SB-injected zebrafish compared to WT (*p* ≤ 0.05, Appendix AC), and TB-injected zebrafish showed an increase of ~38% compared to CTRL-injected zebrafish (*p* ≤ 0.05, Appendix AC). C18 levels increased ~90% in TB compared to WT (*p* ≤ 0.01, Appendix AE) and ~48% compared to CTRL (*p* ≤ 0.05, Appendix AE). C18:1 increased 92% in TB-injected zebrafish compared to CTRL-injected zebrafish (*p* ≤ 0.05, Appendix AF). Very-long-chain fatty acylcarnitines, C22:5 and C22:6, were dramatically affected in CPT2 knockdown larvae. C22:5 levels in TB-injected zebrafish and SB-injected zebrafish showed an increase of ~235% (*p* ≤ 0.01, Appendix AL) and ~258% (*p* ≤ 0.001, Appendix AL), respectively, compared to CTRL-injected zebrafish. C22:6 is increased in SB larvae by ~395% compared to CTRL-injected zebrafish (*p* < 0.0001, Appendix AM). These data further support that CPT2 expression and function in the carnitine shuttle system are disrupted by MOs in the zebrafish model.

### 3.3. Effect of CPT2 Knockdown on Zebrafish Morphology

At 5 dpi, CPT2 knockdown larvae demonstrated abnormal phenotypes as compared to WT and CTRL-injected zebrafish (Figure 3). Pericardial edema, curved tail, abnormal yolk sac, and abdomen shape were measured in TB- and SB-injected zebrafish as compared to WT and CTRL-injected zebrafish. Phenotypic differences were the most prevalent in SB-injected zebrafish compared to WT zebrafish (Figure 3A). Pericardial edema increases ~20% in SB-injected zebrafish compared to WT (*p* < 0.001, Figure 3B) and ~14% compared to CTRL-injected zebrafish (*p* ≤ 0.05, Figure 3B). No significant change in pericardial edema was noted in TB-injected zebrafish fish when compared to WT (~2% increase, *p* = 0.97, Figure 3B) or CTRL-injected zebrafish (~5% increase, *p* = 0.8434, Figure 3B). The percent of larvae with curved tails increased by ~34% in SB-injected zebrafish compared to WT (*p* < 0.0001, Figure 3C) and by ~28% compared to CTRL-injected zebrafish (*p* < 0.001, Figure 3C). There was no significant change in TB-injected zebrafish with curved tails when compared to WT (*p* = 0.43, Figure 3C) or CTRL-injected zebrafish (*p* = 0.94, Figure 3C). The occurrence of an abnormal yolk sac or abdomen was not significantly different between MO-injected zebrafish and control-injected zebrafish (Figure 3D). Abnormal eye and tail development was observed in CPT2 knockdown larvae and additional measurements were conducted to assess body and brain development. Eye length, rump length, and standard length were measured from the lateral side of the larva. Eye length was measured to evaluate eye development, rump length was measured to assess gastrointestinal development, standard length was measured to analyze body development, and eye width was measured from the dorsal view to indirectly examine brain development (Figure 4A). TB-injected zebrafish had reduced rump length by 128.1 ± 42.0 µm (*p* ≤ 0.05, Figure 4C) and reduced eye width by 28.2 ± 10.5 µm (*p* ≤ 0.05, Figure 4E) compared to CTRL-injected zebrafish but were not significantly different from WT larvae (Figure 4B–E). 

In SB-injected zebrafish, eye length was significantly decreased by 42.4 ± 7.8 µm compared to WT (*p* < 0.0001, Figure 4B) and 48.5 ± 7.9 µm compared to CTRL-injected zebrafish (*p* < 0.0001, Figure 4B). Rump length was significantly decreased by 290.1 ± 41.3 µm compared to WT (*p* < 0.0001, Figure 4C) and 316.7 ± 42.3 µm compared to CTRL-injected zebrafish (*p* < 0.0001, Figure 4C). Standard length was significantly decreased to 534.5 ± 69.5 µm compared to WT (*p* < 0.0001, Figure 4D) and 583.3± 71.1 µm compared to CTRL-injected zebrafish (*p* < 0.0001, Figure 4D). Eye width was significantly decreased by 5.3 ± 10.3 µm compared to WT (*p* < 0.0001, Figure 4E) and 52.8 ± 10.5 µm compared to CTRL-injected zebrafish (*p* < 0.0001, Figure 4E). These data indicate that CPT2 knockdown affects the gross development of larvae at 5 dpi.

### 3.4. Effect of CPT2 Knockdown on the Lipid Deposition in Zebrafish

CPT2 knockdown may influence the deposition of lipids and cholesterol in zebrafish larvae. Oil Red O was used to observe neutral lipids and cholesterol esters in 5 dpi larvae (Figure 5). Both TB-injected and SB-injected zebrafish larvae showed an increase in Oil Red O staining when compared to CTRL-injected and WT zebrafish (Figure 5A). Staining intensity was evaluated using Fiji ImageJ by demarcating the brain and yolk sack and measuring pixel intensity (Figure 5B). Oil Red O staining intensity was not significantly different in the brains of TB- and SB-injected zebrafish compared to CTRL-injected and WT zebrafish (*p* > 0.05, Figure 5C). Oil Red O staining intensity in the yolk sac of SB-injected larvae increased ~33% compared to WT (*p* ≤ 0.05, Figure 5D) and ~90% compared to CTRL-injected zebrafish (*p* ≤ 0.0001, Figure 5D). Oil Red O staining intensity of the yolk sac in TB-injected larvae was not significantly different compared to WT or CTRL-injected zebrafish. These data suggest that significant CPT2 knockdown in SB-injected larvae resulted in an inability to effectively utilize and/or mobilize their lipid stores during this early stage of development.

### 3.5. Effect of CPT2 Knockdown on the Developing Nervous System in Zebrafish

Craniofacial abnormalities can occur in forms of human CPT2 deficiency [11]. Since craniofacial development requires proper neural tube closure and neural crest cell migration and may be influenced by abnormal LCFA utilization, the development of craniofacial cartilage in CPT2 knockdown larvae was evaluated compared to WT and CTRL-injected zebrafish. A narrowing of the Mechel’s cartilage and ceratohyal cartilage, as well as a change in the development of ceratobranchials i–v, was observed in some CPT2 knockdown larvae (Figure 6A). Three separate measurements were quantified to evaluate cartilage formation: (i) the distance from Mechel’s cartilage to the ceratohyal cartilage, (ii) the angle of the Mechel’s cartilage, (iii) and the angle of the ceratohyal cartilage (Figure 6B). There were no significant differences in the development of Mechel’s or ceratohyal or ceratobranchial cartilage between CPT2 knockdown and control larvae (*p* > 0.05, Figure 6C–E). These data indicate that CPT2 knockdown does not significantly influence cartilage development as measured by these specific parameters at 5 dpi in this zebrafish model system. 

To evaluate potential changes to gross brain development in CPT2 knockdown larvae, brains were dissected from 5 dpi larvae and immunostained for acetylated tubulin. Whole brain area, forebrain area, forebrain width, midbrain area, optic tectum area, and cerebellum area were measured (Figure 7A). Whole brain area decreased by ~39% in SB compared to WT and ~38% compared to CTRL-injected zebrafish (*p* < 0.0001, Figure 7B). Forebrain area was decreased in SB-injected by ~31% compared to WT (*p* < 0.001, Figure 7C) and ~33% compared to CTRL-injected zebrafish (*p* < 0.0001, Figure 7C). Forebrain length was not significantly different between any conditions (*p* > 0.05, Figure 7D). Midbrain area decreased ~64% in SB-injected compared to WT (*p* < 0.0001, Figure 7E) and ~61% compared to CTRL-injected zebrafish (*p* < 0.0001, Figure 7E). Optic tectum area increased over 250% compared to WT and CTRL-injected zebrafish (*p* < 0.001, Figure 7F). Cerebellum area decreased ~37% in SB-injected compared to WT (*p* < 0.0001, Figure 7G) and 22.6% compared to CTRL-injected zebrafish (*p* ≤ 0.05, Figure 7G). TB-injected zebrafish cerebellar area decreased ~26% compared to WT (*p* < 0.01, Figure 7G). These data provide evidence that CPT2 knockdown significantly disrupted normal brain development in this zebrafish model system.

Since the acetylation of tubulin is a post-translational modification associated with cell development and vesicle transport [34,35], neuronal projections identified by immunostaining of acetylated tubulin were measured (Figure 8). DAPI was used as a nuclear counterstain. The optic tectal regions, outlined in white, were evaluated from the dorsal side of the zebrafish brain (Figure 8A). The optic tectum is homologous to the human superior colliculus that is responsible for the perception of and response to environmental stimuli [36]. The tectum receives afferent inputs from multiple regions of the zebrafish brain including the contralateral eye and the hypothalamus [36]. SB-injected optic tectal regions showed a significant increase in acetylated tubulin by ~84% compared to WT (*p* < 0.0001, Figure 8B) and ~119% compared to CTRL-injected zebrafish (*p* < 0.0001, Figure 8B). Ventrally located, the paraventricular organ posterior part (PVOp) is a part of the hypothalamus, and acetylated tubulin labeling in knockdown fish was unique to that of controls (Figure 8C). The PVOp of the hypothalamus is highly conserved between zebrafish and humans and regulates food intake as well as locomotor activity through afferent input from the dorsal motor nucleus of the vagus nerve [36,37,38]. TB-injected zebrafish showed a significant decrease in mean intensity of the labeling by ~29% compared to WT (*p* ≤ 0.05, Figure 8D) and ~29% compared to CTRL-injected zebrafish (*p* ≤ 0.05, Figure 8D). SB-injected zebrafish showed a significant increase of ~29% compared to WT (*p* ≤ 0.05, Figure 8D) and ~29% compared to CTRL-injected zebrafish (*p* < 0.01, Figure 8D). These data show increased acetylated tubulin labeling in the optic tectal regions and the PVOp in SB-injected fish compared to controls.

Zebrafish midbrain catecholinergic neurons are dopaminergic and are homologous to mammalian midbrain catecholaminergic neurons that are functionally important for motor control, reward, motivation, and cognitive function [39,40]. Tyrosine hydroxylase (TH) is the rate-limiting enzyme of catecholamine biosynthesis, and dysfunctional TH activity is seen in schizophrenic patients [40,41,42]. Fatty acid metabolism has recently been shown to be important for catecholinergic neuronal function and TH activity [40,41,42,43,44], and the proband diagnosed with CPT2 deficiency described in our case study experiences schizophrenic-like symptoms [15]. In zebrafish, TH pathways are important in swimming movement [45]. Anti-TH1 immunofluorescent analyses were performed to determine how CPT2 knockdown affects TH+ catecholaminergic neurons. Zebrafish have two homologs for TH, TH1 and TH2. The TH1 enzyme uses tetrahydrobiopterin and molecular oxygen to convert tyrosine to DOPA, and the homolog TH2 enzyme functions as mammalian tryptophan hydroxylase to convert tryptophan to 5-hydroxy-L-tryptophan (5-HTP) and 4a-hydroxy-BH4 (pterin-4a-carbinoxamine derivative of BH4) during the rate-limiting step of serotonin synthesis [40]. The TH antibody is immunospecific for both TH1 and TH2 and binds to dopaminergic and serotonergic neurons. The hypothalamus and particularly the PVOp were evaluated from the ventral side of the fish for TH immunoreactivity (Figure 8E). Mean staining intensity of the TH+ cells in the hypothalamic region increased ~35% in SB-injected zebrafish compared to WT (*p* < 0.0001, Figure 8F) and ~41% when compared to CTRL-injected zebrafish (*p* < 0.0001, Figure 8F). TH+ cells were counted in the hypothalamus and SB-injected larvae showed a ~12% decrease compared to WT and a ~15% decrease compared to CTRL-injected zebrafish (*p* ≤ 0.05, Figure 8G). Differences in overall brain morphology and immunofluorescent analyses suggest that axonal projections and the expression of enzymes involved in the development and function of catecholaminergic neurons may be affected by CPT2 knockdown.

### 3.6. Effect of CPT2 Knockdown on Behavior and Brain Function

TH+ neuronal neurons and networks are important for swimming movement. Since TH+ immunofluorescence showed significant effects on TH+ neurons and TH+ networks, swimming movement was evaluated. The ViewPoint Zebrabox system (Version 3.22) was used to evaluate whether CPT2 knockdown affected swimming behavior of fish during light stimulation (Figure 9A). The quantization assay evaluated larval swimming speed and duration. Swimming activity count measures the number of times a larva entered a particular activity state. Swimming activity duration measures the amount of time a larva spent in a particular activity state. Activity states were defined as freezing activity, medium activity, and bursting activity. Measurements of behavioral activity for each zebrafish were averaged and the changes in MO-injected zebrafish behavior compared to CTRL-injected and WT zebrafish were analyzed. SB-injected larvae entered the freezing activity state less frequently than WT and CTRL-injected zebrafish by 434.6 ± 97.3 (*p* < 0.0001, Figure 9B) and 627.0 ± 100.7 (*p* < 0.0001, Figure 9B), respectively. Once SB-injected larvae entered the freezing state, SB-injected zebrafish larvae spent significantly more time in the freezing activity state by 60.0 ± 11.3 s compared to WT (*p* < 0.0001, Figure 9E) and 112.0 ± 11.7 s compared to CTRL-injected zebrafish (*p* < 0.0001, Figure 9E). Interestingly, SB-injected fish did not enter, or spend as much time in, the medium activity state as did WT or CTRL-injected zebrafish (*p* < 0.0001, Figure 9C,F) (*p* < 0.0001, Figure 9F). SB-injected zebrafish showed a significant decrease in the number of times they entered the bursting state by 58.1 ± 11.1 counts compared to WT (*p* < 0.0001, Figure 9D) and 78.5 ± 11.5 counts compared to CTRL-injected zebrafish (*p* < 0.0001, Figure 9D). SB-injected zebrafish larvae spent significantly less time in the bursting activity state, demonstrating a reduction of 13.6 ± 2.7 s compared to WT (*p* < 0.0001, Figure 9G) and 16.6 ± 2.8 s compared to CTRL-injected zebrafish (*p* < 0.0001, Figure 9G). Overall, SB-injected zebrafish showed a significant decrease in activity compared to WT and CTRL-injected zebrafish. TB-injected zebrafish demonstrated more subtle behavioral changes from WT and CTRL-injected zebrafish.

Behavioral tracking assays were used to measure the rate and distance of swimming during light stimulation. Zebrabox tracking software (Version 3.22) races zebrafish movement and records small movement as a red line and large movements as green lines. Swimming near the edges and decreased movement toward the center of the well indicate thigmotaxic avoidance behavior and are well documented in zebrafish [46,47]. CPT2 knockdown larvae demonstrated increased thigmotaxis as shown by the decreased number of wells with red tracking lines in the center of wells as compared to WT, CTRL-, and TB-injected zebrafish (Figure 10A). The duration of time spent in a small movement state was significantly reduced in SB-injected larvae by 173.3 ± 43.8 s compared to WT and by 204.3 ± 47.4 s compared to CTRL-injected zebrafish (*p* < 0.001, Figure 10B). The duration of time spent in a large movement state was significantly reduced in SB-injected zebrafish compared to WT (*p* < 0.001, Appendix A) and CTRL-injected zebrafish (*p* < 0.01, Figure 10C). SB-injected zebrafish traveled 16,684 ± 2229 µm less than WT (*p* < 0.0001, Figure 10D) and 11,722 ± 2411 µm less than CTRL-injected zebrafish (*p* < 0.0001, Figure 10D) when in the large movement state. No significant differences were noted between conditions for inactivity count, duration, and distance; small movement count and distance; and large movement count in the tracking assay (Appendix A–F). These data suggest that CPT deficiency results in a reduction in locomotion and ability to change movement states concurrent with an increase in avoidance behaviors. 

### 3.7. Effect of CPT2 Knockdown on Neural Network Activity

Since seizures have been associated with more severe forms of CPT2 deficiency, and our proband with CPT2 deficiency presented with seizures during infancy [15], neural network activity of CTRL-injected and SB-injected 5 dpi larvae (Figure 11 and Appendix A) was monitored for seizure susceptibility compared to controls. MED64 electrical traces of raw and filtered recordings over the 30 min period were able to detect normal activity during the baseline, followed by an increase in electrical activity after the addition of PTZ, the GABA_A_ receptor antagonist. Following PTZ perfusion, the perfusion of AP5 and CNQX reduced neuronal activity. SB-injected zebrafish demonstrated increased baseline neuronal activity following the addition of PTZ compared to CTRL-injected zebrafish (Appendix A). The spike rate was determined for each larva. During aCSF perfusion, SB-injected zebrafish larvae had a significant spike rate increase of 2.5 ± 1.3 Hz compared to CTRL-injected zebrafish (*p* < 0.0001, Figure 11E). During the addition of PTZ, SB-injected larvae showed a significant increase in network activity of 4.0 ± 2.0 Hz when compared to CTRL-injected zebrafish (*p* < 0.0001, Figure 11E). During the perfusion of AP5 and CNQX, SB-injected zebrafish showed an increase in network activity of 1.7 ± 0.8 Hz when compared to CTRL-injected zebrafish (*p* < 0.001, Figure 11E). These data indicate that CPT2 knockdown affected electrical activity and induced a seizure-like phenotype when compared to CTRL-injected zebrafish larvae.

Increased glutamate release can result in increased NMDA receptor expression [48,49,50]. Because electrical activity in SB-injected zebrafish was significantly increased and because the increase was reduced by antagonists of glutamate neuronal signaling, the expression of the zebrafish glutamatergic receptor, *glutamate ionotropic receptor NMDA type subunit 1a* gene (*Grin1a*), was evaluated (Figure 11F). *Grin1a* is homologous to human *GRIN1* and encodes the NMDA receptor subunit NR1 [48,49,50]. *Grin1a* expression increased in TB-injected larvae by ~48% compared to CTRL-injected larvae; however, this increase was not significant (*p* = 0.1397, Figure 11F). SB-injected larvae showed a significant increase in *Grin1a* expression by ~79% compared to CTRL-injected zebrafish larvae (*p* ≤ 0.05, Figure 11F). These data suggest that there is an association with increased seizure-like activity and NMDA receptor expression in CPT2 knockdown zebrafish.

### 3.8. Effect of CPT2 Knockdown on Mitochondrial and Neuronal Gene Expression

The zebrafish CPT2 knockdown model system presents structural and functional changes, suggesting that disrupted lipid metabolism influences gene expression broadly. CPT2 knockdown zebrafish were evaluated for changes in gene expression related to mitochondrial function, neurotransmitter function and signaling, and schizophrenia-like disease states at the early larvae stage. RT-qPCR was used to assess changes in homologous gene expression in zebrafish in SB-injected and TB-injected zebrafish compared to CTRL-injected zebrafish because CTRL-injected and WT zebrafish showed similar morphology and behaviors. 

CPT2 is expressed in the mitochondria so the effect of CPT2 knockdown on mitochondrial gene expression was evaluated. Mitochondrial genes that were evaluated were *Transcriptional factor A mitochondrial* (*Tfam*), *Pparg coactivator 1 alpha* (*Ppargclα*), *fatty acid binding protein 3* (*Fabp3*), and *leucine-rich pentatricopeptide repeat containing* (*Lrpprc*). All mitochondrial genes were significantly upregulated in SB-injected zebrafish compared to CTRL- and TB-injected zebrafish (Figure 12A). In SB-injected zebrafish, *Tfam* expression increased ~11-fold compared to CTRL-injected (*p* < 0.001, Figure 12A) and ~4-fold compared to TB-injected zebrafish (*p* < 0.0001, Figure 12A). *Ppargclα* expression increased ~6-fold in SB-injected zebrafish compared to CTRL-injected and ~4-fold compared to TB-injected zebrafish larvae (*p* < 0.0001, Figure 12A). *Fabp3* expression in SB-injected zebrafish increased ~4-fold compared to CTRL-injected and ~3-fold compared to TB-injected zebrafish larvae (*p* < 0.0001, Figure 12A). The expression of *Ppargclα* and *Fabp3* did not significantly change in TB-injected zebrafish compared to CTRL-injected zebrafish (Figure 12A).

Since a change in tyrosine hydroxylase (TH) immunolabeling was demonstrated in SB-injected larvae, the expression of homologous TH and dopaminergic genes was investigated. Dopaminergic genes of interest that were evaluated by RT-qPCR were dopamine receptor genes *D1b* (*Drd1b*), *D2a* (*Drd2a*), *D3* (*Drd3*), and *D4a* (*Drd4a*); the dopamine transporter *Solute carrier family 6 member* 3 (Slc6a3); and dopamine synthesis genes *Th1* and *Th2*. *Drd1b* expression increased ~5-fold in SB-injected zebrafish compared to CTRL-injected zebrafish and ~4-fold compared to TB-injected larvae (*p* < 0.0001, Figure 12B). No significant change in expression was noted for *Drd2a* between CPT2 knockdown larvae and CTRL-injected larvae. *Drd3* expression increased ~7-fold in SB-injected zebrafish compared to CTRL-injected zebrafish and ~5-fold compared to TB-injected zebrafish (*p* < 0.0001, Figure 12B). *Drd4a* expression increased ~4-fold in SB-injected zebrafish compared to CTRL-injected zebrafish (*p* ≤ 0.05, Figure 12B). These data show an overall increase in the expression of genes encoding dopamine receptors in SB-injected CPT2 knockdown zebrafish. No significant changes in *Slc6a3* expression were detected (Figure 12B). There are two tyrosine hydroxylase homologs in zebrafish, *Th1* and *Th2*. TH1 is involved in the synthesis of dopamine and TH2 is involved in dopamine and serotonin synthesis in zebrafish [42]. *Th1* expression increased ~7-fold in SB-injected zebrafish compared to CTRL-injected zebrafish and increased ~5-fold compared to TB-injected zebrafish (*p* < 0.0001, Figure 12B). *Th2* expression increased ~2-fold in SB-injected zebrafish compared to CTRL-injected zebrafish (*p* ≤ 0.05, Figure 12B). These data indicate a change in dopamine receptor and TH gene expression in SB-injected CPT2 knockdown larvae.

To investigate how CPT2 knockdown affected serotonergic, cholinergic, and glutamatergic neuronal gene expression, the translation of *hydroxytryptamine receptor 1b* (*Htr1b*), *cholinergic receptor muscarinic 2a* (*Chrm2a*), and *Vglut* genes was evaluated. *Htr1b* and *Chrm2a* encode serotonergic and muscarinic cholinergic receptors, respectively. HTR1B is a G-coupled receptor bound by serotonin. The expression of *Htr1b* in SB-injected zebrafish was increased ~5-fold compared to CTRL-injected zebrafish (*p* < 0.0001, Figure 12C) and ~3-fold compared to TB-injected zebrafish (*p* < 0.001, Figure 12C). SB-injected zebrafish showed an ~8-fold increase in *Chrm2a* expression compared to CTRL-injected zebrafish (*p* < 0.01, Figure 12C) and a ~4-fold increase compared to TB-injected zebrafish (*p* < 0.01, Figure 12C). *Vesicular Glutamate Transporter 1* (*Vglut1*) and *Vesicular Glutamate Transporter 2* (*Vglut2*) are glutamate transporter genes expressed in glutamatergic neurons. These protein-coding genes are involved with the vesicular packaging of glutamate for transport. *Vglut1* expression in SB-injected zebrafish decreased ~0.1-fold compared to CTRL (*p* < 0.01, Figure 12C) and TB-injected zebrafish (*p* < 0.001, Figure 12C). *Vglut2* expression in SB-injected zebrafish increased ~2-fold compared to CTRL-injected zebrafish (*p* < 0.01, Figure 12C). These data show that CPT2 knockdown during early development alters normal serotonergic, cholinergic, and glutamatergic gene expression in the zebrafish vertebrate model system.

To further analyze the relationship, a potential link between CPT2 deficiency and the expression of schizophrenia-related genes, *α-synuclein interacting protein* (*Sncaip*) and *disrupted in schizophrenia 1* (*Disc1*), was evaluated. SNCAIP is an α-synuclein interacting protein and is involved in synaptic plasticity because SNCAIP participates with α-synuclein to maintain synaptic vesicle pools and regulate neurotransmitter release from the presynaptic neuron [51]. *Sncaip* expression increased ~5-fold in SB-injected zebrafish compared to CTRL-injected zebrafish (*p* < 0.001, Figure 12D) and ~4-fold compared to TB-injected zebrafish (*p* < 0.0001, Figure 12D). DISC1 is associated with synaptic pruning during critical periods of development [52,53,54]. The expression of *Disc1* in SB-injected zebrafish increased ~5-fold compared to CTRL-injected zebrafish (*p* < 0.001, Figure 12D) and ~4-fold compared to TB-injected zebrafish (*p* < 0.0001). These data indicate a change in *Sncaip* and a key schizophrenia marker, *Disc1*, in CPT2 knockdown larvae. 

Recently, α-synuclein aggregation has been shown to increase in cells and cerebrospinal fluid [55] and decrease in the blood serum [56,57] of people with dopamine-related disease states including schizophrenia. To evaluate α-synuclein expression in the zebrafish CPT2 knockdown model system, Western blotting was used. An increased expression of phosphorylated and unphosphorylated α-synuclein protein in SB-injected zebrafish compared to CTRL-injected zebrafish (Figure 12E) was observed. The quantification of all Western blots showed a ~3-fold increase in the protein expression of α-synuclein (*p* ≤ 0.01, Figure 12F) and ~2-fold increase in phosphorylated α-synuclein (*p* ≤ 0.05, Figure 12G) in SB-injected zebrafish compared to CTRL-injected zebrafish. There was no significant difference between unphosphorylated and phosphorylated α-synuclein levels in SB-injected zebrafish. In addition, there was not a significant change in unphosphorylated or phosphorylated α-synuclein in TB-injected zebrafish compared to CTRL-injected zebrafish (Figure 12E–G). These data provide evidence that α-synuclein protein is significantly increased in cells and tissues of CPT2 knockdown zebrafish. A tabulated summary of the Results Section can be found in Appendix A.

## 4. Discussion

Previously, we reported that human CPT2 deficiency in a male proband was comorbid with seizures in early development and schizophrenia in early adulthood [15]. Normal CPT2 function is required for carnitine on a fatty acid to be replaced with a coenzyme and acylcarnitine so that long-chain fatty acids can be effectively utilized for β-oxidation. CPT2 deficiencies in humans result in a variety of clinical outcomes depending on the severity of the CPT2 mutation; however, mild-to-moderate CPT2 deficiency had not previously been linked to the development of schizophrenia [10,11,12,13,14,15]. The goal of this study was to develop an accessible vertebrate model system, which is highly homologous to humans, to begin to investigate the effects of CPT2 deficiency on early neurodevelopmental events that may help explain the putative association to schizophrenia susceptibility. As a proof of concept, the CPT2 knockdown model employed both translation and splice blocking MOs to knockdown, but not eliminate, CPT2 function at the single-cell stage of zebrafish development. MOs are commonly used in zebrafish and are an effective tool for gene knockdown studies [19,20,21,22,23,24]. As demonstrated by PCR and Western blotting, MO oligonucleotides targeting *cpt2* expression in zebrafish reduced CPT2 protein expression (Figure 1 and Figure 2). TB MOs and SB MOs provide a means to compare mild-to-moderate CPT2 knockdown to control zebrafish in these and future studies. Future rescue experiments will evaluate the ability of injected *cpt2* mRNA to restore zebrafish knockdown phenotypes. NMD-blocking strategies could be used to further validate SB knockdown.

Deficits in the carnitine system may result in abnormal acylcarnitine long-chain fatty acids and lipid deposition and utilization [13,14]. LC-MS/MS analyses determined that total acylcarnitine levels were significantly increased in CPT2 knockdown larval zebrafish compared to WT zebrafish (Figure 2D). While the SB morpholino was more effective at knocking down CPT2 expression, and generally had more significant phenotypes, the TB morphants demonstrated larger changes in total acylcarnitine levels (Figure 2D). Under physiologic conditions, the oxidation of long- and medium-chain fatty acids is primarily handled by the mitochondrial β-oxidation system, with only minimal contribution from the peroxisomal system [9,14]. Partial CPT2 expression as seen in TB-injected zebrafish could result in a greater accumulation of acylcarnitine species compared to SB-injected zebrafish because the signaling to oxidize fatty acids in peroxisomes or by ω-oxidation may be incompletely activated due to the remaining CPT2 function in mitochondria. The accumulation of some acylcarnitine species in TB- and SB-injected zebrafish compared to CTRL-injected and WT zebrafish was interesting (Appendix A). Patients with CPT2 deficiency have been shown to have increased levels of acylcarnitine C14, C16, C16:1, C18, C18:1, and C18:2 in blood serum [13,14,58,59,60]. The accumulation of C16 and the ratio of C16 + C18:1/C2 have previously been used to identify CPT2 deficiency. Meta-analyses found significant increases in acylcarnitines with at least 12 or more carbons in people with long-chain fatty acid oxidation disorders and CPT2 deficiency [61,62]. CPT2 knockdown zebrafish are similar as they demonstrate significant increases in larger-carbon-chain acylcarnitine species (Appendix A). Like patients with CPT2 deficiency, C16, C18, and C18:1 acylcarnitines were significantly increased in TB-injected when compared to WT and CTRL-injected zebrafish. SB-injected zebrafish had a significant increase in acylcarnitine levels of C16 when compared to WT while C18 and C18:1 increased moderately albeit not significantly in SB-injected zebrafish compared to WT and CTRL-injected zebrafish (Appendix A). Levels of acylcarnitine long-chain fatty acids such as C22:5 and C22:6 were significantly increased in SB-injected as compared to WT and CTRL-injected zebrafish (Appendix A). Some acylcarnitine species were challenging to evaluate due to sample variability and very low abundance such as C20:2, C20:4, and C20:5 species (Appendix A). When making comparisons between human studies and the zebrafish model system presented here, it is important to note that there may be species-specific differences. Further, acylcarnitine levels in CPT2 deficiencies vary widely between forms of the disease and acylcarnitine levels in serum, blood spots, and tissue differ [62,63]. Typically, blood serum is collected from humans for an acylcarnitine analysis, while, here, whole body tissues were prepared from zebrafish for LC-MS/MS analyses. Despite these potential differences, in each MO-injected zebrafish, CPT2 deficiency was associated with an increase in acylcarnitine long-chain fatty acids including increased levels of C16. Interestingly, recent studies suggest that acylcarnitine levels are altered in autistic and schizophrenic patients and that carnitine species are influenced by antipsychotic medication [64]. The characterization of acylcarnitine species may yield important insight into how lipid and metabolic processes contribute to underlying mechanisms of heterogeneous neurodevelopmental and neurocognitive disorders. Future experiments may examine effects of drug treatment or supplementation on acylcarnitine species accumulation, taking advantage of the zebrafish high-throughput capabilities.

SB-injected zebrafish exhibited significant changes in body, brain, and eye development compared to CTRL-injected and WT zebrafish (Figure 3 and Figure 4). In addition to the reduction in overall body size, significant increases in pericardial edema and abnormal tail structure were observed in CPT2 knockdown zebrafish (Figure 3A–C). The presentation of reduced body size and a significant increase in curved tails in CPT2 knockdown zebrafish is likely associated with improper muscle development [58]. CPT2 deficiency can affect muscle development and function in humans. Curved tails could also arise from neural tube defects in zebrafish, and the lethal neonatal form of CPT2 deficiency is associated with defects in neurulation [58]. Cardiomyopathy is a common symptom associated with the severe infantile hepatocardiomuscular form of CPT2 deficiency [12,13,14]. SB-injected zebrafish presented with enlarged ventricles and the accumulation of fluid during cardiac development when compared to WT and CTRL-injected zebrafish. In TB-injected zebrafish and SB-injected zebrafish, eye size and gastrointestinal development were reduced compared to WT and CTRL-injected zebrafish (Figure 4). The phenotypic characterization of CPT2 knockdown with Oil Red O staining indicated that CPT2 knockdown significantly increased lipid deposition in SB-injected zebrafish compared to WT and CTRL-injected zebrafish (Figure 5A–D). Zebrafish have a yolk sac for approximately 5–6 days after fertilization to provide them with a source of proteins, lipids, and other micronutrients that are used to produce energy and support cellular processes during early development [65]. The increase in lipid deposition in the yolk sac of SB-injected zebrafish larvae suggests that lipids are not being utilized properly in zebrafish where CPT2 is knocked down. Here, these data suggest that the zebrafish CPT2 knockdown model system had an inefficient utilization of lipids and therefore can be used to evaluate coincident changes in brain structure and function.

Since severe lethal neonatal forms of CPT2 deficiency in humans result in abnormal craniofacial development and a decrease in cranial development and head size [12,13,14], craniofacial development in CPT2 knockdown fish using Alcian blue staining was assessed. In the zebrafish CPT2 knockdown model, abnormal cranial development was not observed (Figure 6). These data suggest that this CPT2 knockdown model system is well suited to investigate mild-to-moderate forms of CPT2 deficiency rather than the lethal neonatal form and severe infantile hepatocardiomuscular form of CPT2 deficiency where abnormal cranial development occurs.

While craniofacial development was not significantly different in SB-injected zebrafish compared to controls, abnormal brain development was exhibited. SB-injected zebrafish larvae brains had decreased forebrain, midbrain, and cerebellar areas, and increased optic tectal areas, compared to WT and CTRL-injected zebrafish larvae (Figure 7). These data support the potential importance of the carnitine system fatty acid oxidation during early brain development for proper brain formation. Abnormal acylcarnitine levels and β-oxidation of LCFA in the brain can affect the composition of cellular membranes, expression of genes and proteins, function of mitochondria, regulation of antioxidant activity, frequency of neurotransmission, and prevalence of programmed cell death [8,59]. SB-injected zebrafish also demonstrated abnormal axonal outgrowth and TH immunoreactivity (Figure 8). Highly branched and disorganized axonal projections at early stages of neural development may be due to inappropriate axonal guidance, aberrant neurotransmitter release, increased neural network activity, and/or a reduction in programmed cell death. The brain regions of SB-injected zebrafish that showed the most change in acetylated tubulin immunoreactivity were the optic tectum and paraventricular organ posterior (PVOp) region. The optic tectum aids in the perception of visual stimuli and receives afferent inputs from multiple regions including the contralateral eye and the hypothalamus [36]. The paraventricular organ posterior (PVOp) houses dopaminergic neurons and is associated with a wide range of hypothalamic functions such as sodium uptake, glucose metabolism, cardiovascular function, motor function, and gastrointestinal and respiratory activities [38,39]. The PVOp is a functional homolog to the substantia nigra in mammals. The substantia nigra houses dopaminergic neurons and affects movement control, cognitive executive functions, and emotional limbic activity. Brains of schizophrenic patients demonstrate structural change in, and enhanced electrical excitability of, the substantia nigra [41,43,66,67].

Immunofluorescent analyses of TH+ cells demonstrated an increase in TH+ intensity but a decrease in the number of TH+ neurons in SB-injected zebrafish compared to controls (Figure 8). TH immunoreactivity may label both dopaminergic and serotoninergic neurons because the TH antibody recognizes both TH1 and TH2 in zebrafish [41,42,46]. Increased tyrosine hydroxylase mRNA expression, dopamine synthesis, and serotonin production are seen in individuals with schizophrenia [68,69,70,71]. Serotonin and dopamine activity modulators are used clinically to treat dopamine-related disease states such as schizophrenia [64]. Specific single-nucleotide polymorphisms in the promoter region of serotonergic receptor *HTR1B* are associated with schizophrenia and other psychiatric disorders [69,70]. *Chrm2a* is a cholinergic muscarinic receptor gene and is homologous to *CHRM2* in humans. *Chrm2a* and *Chrm2* encode receptors involved in synaptic plasticity and neuronal excitability [72]. In the zebrafish CPT2 knockdown model, significant increases in *Th1* and *Th2*, *Hrt1b*, and *Chrm2a* were measured in SB-injected zebrafish compared to control zebrafish (Figure 12), suggesting that CPT2 deficiency may influence catecholamine and serotonergic neuronal networks. 

The abnormal axonal outgrowth and generation of TH+ neurons are likely to underlie the abnormal swimming behaviors and increased thigmotaxis of SB-injected zebrafish (Figure 9 and Figure 10). TH activity in the nigrostriatal system controls motor functions, and the overexpression of TH could cause a change in motor functioning in CPT2 knockdown larvae. Thigmotaxis is an indicator of an anxiety-like state where an animal avoids open spaces [47]. Zebrafish at 5 dpi exhibit thigmotaxis under specific conditions [73]. While CPT2 deficiency is not commonly associated with increased anxiety, our published case study documents a proband with CPT2 deficiency, anxiety, and schizophrenia [15]. Moderate CPT2 deficiency is commonly associated with seizures in early development [12,13,14]. Increased neural network activity in SB-injected zebrafish larvae was observed during baseline recordings and after the addition of PTZ in the seizure assay compared to CTRL-injected zebrafish (Figure 11 and Appendix A). Interestingly, increased cell proliferation in the optic tectum of an epilepsy zebrafish model has been previously described [74,75]. Future studies could investigate neuronal and glial proliferation and cell death in MO knockdown zebrafish. CPT2 deficiency may cause a hyperexcitable state due to increased cell proliferation and axonal outgrowth in the optic tectum like that shown in epileptic zebrafish models. A lipid-rich ketogenic diet and carnitine supplementation can be used to treat drug-resistant epileptic seizures, suggesting an important role for lipid metabolism and the carnitine system in regulating the electrical activity of neural networks [76,77]. Recently, increased cortical excitability and glutamatergic activity were identified in early stages of the disease progression of schizophrenia [78,79,80,81]. In addition, NMDA receptor activation underlies seizure activity, and NMDA receptor modulators are used to treat epilepsy [82]. RT-qPCR analyses of SB-injected larvae evaluated the coding gene for NMDA receptor NR1, *Grin1a.* The gene expression of transporters for glutamate was also evaluated. Vglut1 expression was significantly decreased and Vglut2 was significantly increased in SB-injected zebrafish compared to controls (Figure 12C). The regulation of both *Vglut1* and *Vglut2* gene expression is associated with the modification of synaptic function [83,84]. Changes in *Vglut1* and *Vglut2* gene expression, due to CPT2 deficiency, may contribute to the abnormal neural network activity and behaviors of CPT2 knockdown zebrafish. The overexpression of *Grin1a* and NR1 can result in a longer latency of seizure activity [50]. In combination, the immunofluorescent data and seizure assays suggest that CPT2 knockdown disrupts the activity pattern of neural networks in the optic tectum and hypothalamic regions. Altered brain oscillations in the default mode network of the hypothalamus are associated with cognitive defects such as mind-wandering symptoms in schizophrenia [39,85]. In the SB-injected CPT2 knockdown model, changes in neural network activity are seen at larval stages of development as more complex behaviors are emerging.

*Tfam* levels affect overall mitochondrial homeostasis and are associated with neurodegenerative disorders like Alzheimer’s and Parkinson’s disease [86]. *Ppargclα* expression is important in mitochondrial biogenesis and fatty acid metabolism. *Ppargclα* expression is stimulated by catecholamines and may contribute to cell signaling underlying neurodegeneration in the substantia nigra [87]. Increased *Fabp3* expression can affect the composition of membranes and increase membrane lipid remodeling [88].FABP3, a mitochondrial-associated protein that aids in lipid transport and is upregulated in SB-injected zebrafish larvae (Figure 12A), also binds α-synuclein and is found in α-synuclein oligomers that are toxic to neurons in synucleopathies [89,90]. *Lrpprc* encodes protein important for a variety of RNA functions in cells and is thought to be involved in the regulation of mitochondrial gene expression. The abnormal expression of *Lrpprc* is associated with increased apoptosis and several disease states including neurodegenerative diseases [91]. Abnormal mitochondrial gene expression and mitochondrial gene mutations are seen in a variety of neurocognitive disorders including schizophrenia [92]. Assessments of a CPT2 knockdown model for mitochondrial function would be interesting future experiments to determine the relevance of changes in *Tfam, Ppargclα, Fabp3,* and *Lrpprc* transcription. 

Immunofluorescent and behavioral studies suggested that CPT2 knockdown influenced dopaminergic networks in early developing zebrafish. RT-qPCR showed that the expression of dopamine receptors, encoded by *Drd* genes, that are important for learning, memory, impulse control, attention, and locomotion [93] is affected by CPT2 knockdown. *Drd1b*, *Drd3*, and *Drd4a* gene expression increased in SB-injected zebrafish compared to CTRL-injected zebrafish (Figure 12B). The expression of the dopamine transporter gene, *Slc6a3*, was not significantly different in SB-injected larvae compared to CTRL-injected larvae (Figure 12B). The dopamine hypothesis proposes that there is a hypodopaminergic state in prefrontal cortical neurons and a hyperdopaminergic state in neurons of the striatum in patients with schizophrenia [41]. These studies show an increase in TH+ immunoreactivity in homologous structures to the striatum in zebrafish (Figure 8) and an increase in the dopaminergic receptor (Figure 12B). Interestingly, genes considered to be markers for schizophrenia and other synucleinopathies were upregulated in CPT2 knockdown zebrafish larvae (Figure 12D). In SB-injected CPT2 knockdown zebrafish, both *Sncaip* and *Disc1* gene expression was increased (Figure 12D). The synuclein alpha interacting protein, synphillin-1, encoded by *Sncaip*, aids in the assembly of the SNARE complex and clustering of synaptic vesicles and participates in synaptic vesicle exocytosis [88]. Normally, α-synuclein acts at the synaptic vesicle membrane to bind VAMP2/synaptobrevin 2 and assist in assembly of the SNARE complex [51]. Abnormal forms or levels of expression of α-synuclein or interacting proteins can induce the assembly of α-synuclein into large oligomers or fibrils that cause neurotoxicity caused by mitochondrial and synaptic dysfunction, inflammation, and apoptosis [89,90,91,92]. Interestingly, lipid production and deposition can influence the expression of synuclein alpha interacting proteins and α-synuclein. Lipid interactions at the membrane can modulate α-synuclein toxicity, and the dysregulation of lipid metabolism may exacerbate synucleinopathies [94,95,96,97]. The investigation of α-synuclein expression in CPT2 knockdown larvae showed that both phosphorylated and unphosphorylated α-synuclein were increased in SB-injected zebrafish compared to CTRL-injected zebrafish (Figure 12E–G). The disrupted-in-schizophrenia 1, *Disc1*, gene encodes a multifunctional scaffold protein. DISC1 is important for intracellular signaling associated with β-catenin regulation, increased dopamine release, dopamine receptor expression, and dopamine transporter function [52]. The misexpression of *Disc1* is associated with an increased risk of mental illness, and polymorphisms of *Disc1* are known genetic risk factors for several psychiatric disorders including schizophrenia [53,54,55]. In CPT2 knockdown fish, the overexpression of the *Disc1* gene and increased axonal immunofluorescence measured by acetylated tubulin at early stages of larval development were observed. CPT2 knockdown studies suggest that the zebrafish model system will be useful for continued studies investigating how metabolic disorders influence gene expression and protein function at early stages of brain development and contribute to neurodegenerative disease states later in life. 

Loss of function mutations in critical enzymes for fatty acid β-oxidation can lead to disrupted brain and body development and nervous system dysfunction. Environmental factors such as the exposure to pesticides and herbicides alter enzyme function and metabolism and can lead to dramatic effects on brain development [98,99]. Toxic effects of environmental factors alter mitochondrial gene expression, metabolic function, and brain structure and function in fish and rodents like the SB zebrafish knockdowns and in our proband. When environmental exposure to toxins coincides with genetic vulnerability, more significant effects on brain development and function are likely and the risk of developing neurological disorders increases.

Our previous work documented a case study of CPT2 deficiency comorbid with early childhood seizures and the development of schizophrenia later in life [15]. These studies highlight how the vertebrate zebrafish model system can model mild-to-moderate CPT2 deficiency. Future work using CRISPR technology to evaluate patient-specific mutations such as that identified in the proband [15] will be used to improve our understanding of how metabolic function and fatty acid β-oxidation contribute to normal neural networking and whether deficiencies in CPT2 are likely to cause dysregulated gene expression associated with neurological disorders such as seizure and dopaminergic disease states such as schizophrenia. The high-throughput capabilities of the zebrafish knockdown model systems may allow for the evaluation of potential preventative or therapeutic strategies for putative and patient-specific mutations.

## Figures and Tables

**Figure 1 biomolecules-14-00914-f001:**
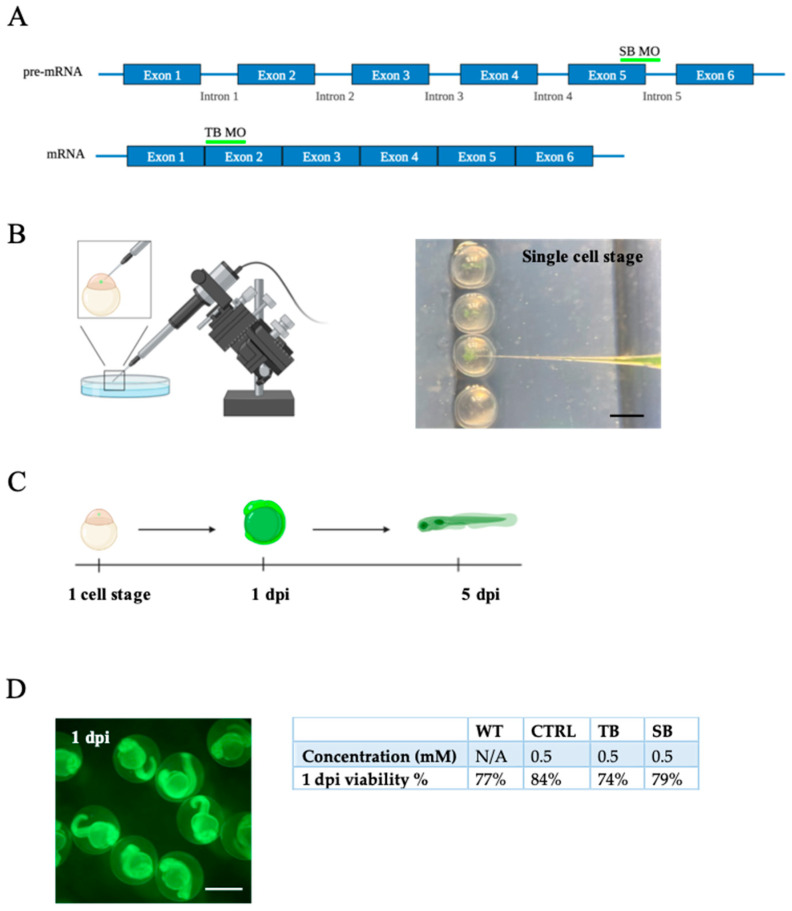
Generation of vertebrate CPT2 knockdown model system in zebrafish. (**A**). Morpholino binding site for splice blocking (SB) and translation blocking (TB) MOs on pre-mRNA and mature mRNA, respectively. (**B**). Graphic of embryos microinjected with fluorescein-labeled morpholino oligonucleotide (MO) at single-cell stage using Narishige injector. Phase bright image of embryos being microinjected at the single-cell stage. (**C**). Timeline CPT2 knockdown model system following MO injection. At 1 dpi, embryos were screened for fluorescein. Viable embryos that were positive for fluorescein (fluorescein+) were raised to 5 dpi for all experiments to evaluate larval development. (**D**). Image of fluorescein+ embryos at 1 dpi. Percent viability of fluorescein+ embryos at 1 dpi compared to uninjected WT zebrafish. N = 15–48 fish per condition for each of three trials. WT = wildtype zebrafish; CTRL = control, scrambled MO-injected zebrafish; TB = translation blocking MO-injected zebrafish; and SB = splice blocking MO-injected zebrafish. (**A**–**C**). Images were designed using BioRender.com (Toronto, ON, Canada). Scale bar = 0.5 mm.

**Figure 2 biomolecules-14-00914-f002:**
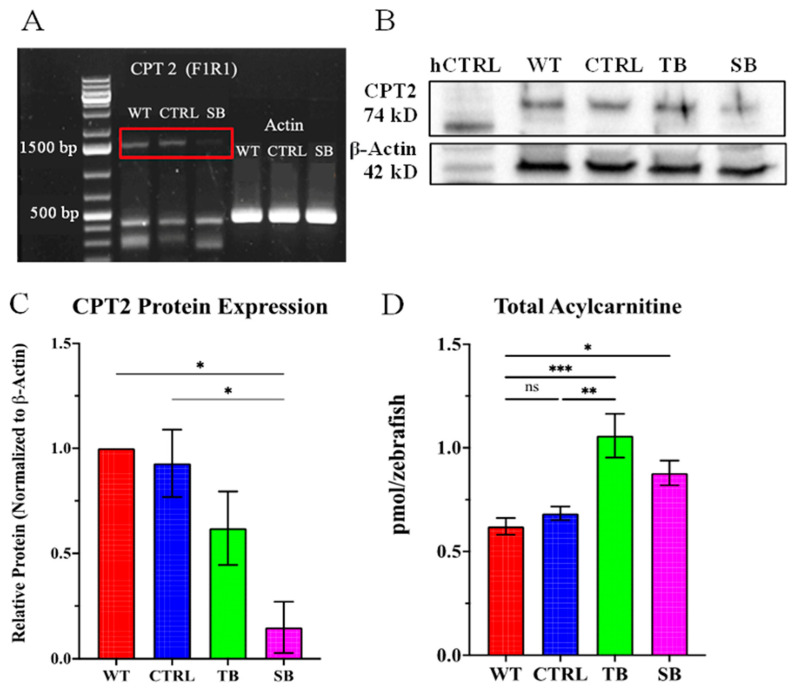
Evaluation of morpholino CPT2 knockdown. (**A**). PCR amplification of full-length *cpt2* run on 1% agarose gel. Actin was used as reference and control. N = 30 fish per condition. Red box indicates the expected *cpt2* PCR product. (**B**). Representative Western blot for CPT2 expression in control and MO knockdown zebrafish. Human CPT2 lysate (hCPT2) was used as positive control. Actin was used as protein loading control. (**C**). CPT2 protein expression in MO knockdown zebrafish relative to WT. Protein levels were normalized to loading control β-actin. N = 200 fish per condition for each trial. Error bars = standard error of difference of 0.19 as determined by ordinary one-way ANOVA. (**D**). Total fatty-acylcarnitine expression evaluated using LC-MS/MS. N = 30 fish per condition for each of five trials. Error bars = SEM. (**A**–**D**). WT = wildtype zebrafish; CTRL = control, scrambled MO-injected zebrafish; TB = translation blocking MO-injected zebrafish; SB = splice blocking MO-injected zebrafish. * *p* ≤ 0.05, ** *p* ≤ 0.01, *** *p* ≤ 0.001, ns = not significant.

**Figure 3 biomolecules-14-00914-f003:**
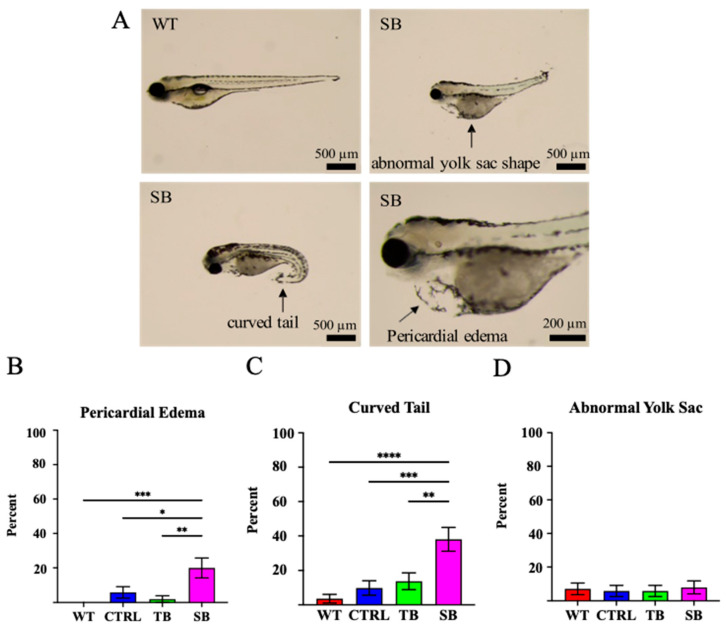
Phenotypic assessment of CPT2 knockdown larvae. (**A**). CPT2 knockdown larvae at 5 dpi presented with several predominant phenotypic differences such as pericardial edema, curved tail, and abnormal yolk sac shape. Representative images of SB-injected phenotypes (SB) and wildtype (WT) comparison are shown. Scale bar = 500 µm and 200 µm, respectively. (**B**–**D**). Percent of qualitative phenotypic differences was compared between conditions. (**A**–**D**). WT = wildtype zebrafish; CTRL = control, scrambled MO-injected zebrafish; TB = translation blocking MO-injected zebrafish; SB = splice blocking MO-injected zebrafish. N = 50 fish per condition. * *p* ≤ 0.05, ** *p* ≤ 0.01, *** *p* ≤ 0.001, **** *p* ≤ 0.0001. Error bars = SEM.

**Figure 4 biomolecules-14-00914-f004:**
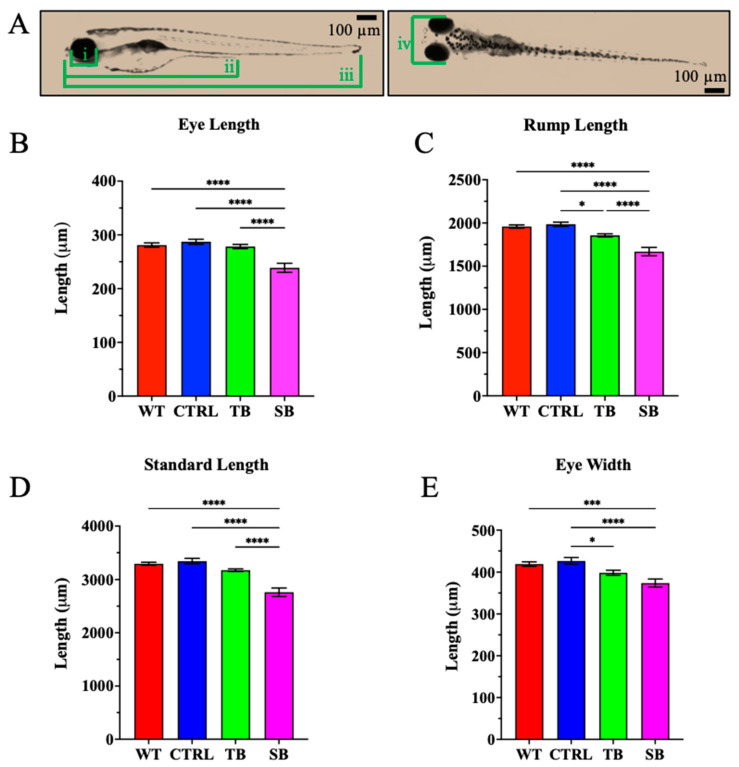
Body development in CPT2 knockdown larvae. (**A**). Measurements of head and body development were obtained using four consistent measurements. (i) Eye length, (ii) rump length, (iii) standard length, (iv) eye width. Scale bar = 100 µm. (**B**–**E**). Measurements for eye length, rump length, standard length, and eye width. (**B**–**E**). WT = wildtype; CTRL = control, scrambled MO-injected zebrafish; TB = translation blocking MO-injected zebrafish; SB = splice blocking MO-injected zebrafish. N = 50 fish per condition. * *p* ≤ 0.05, *** *p* ≤ 0.001, **** *p* ≤ 0.0001. Error bars = SEM.

**Figure 5 biomolecules-14-00914-f005:**
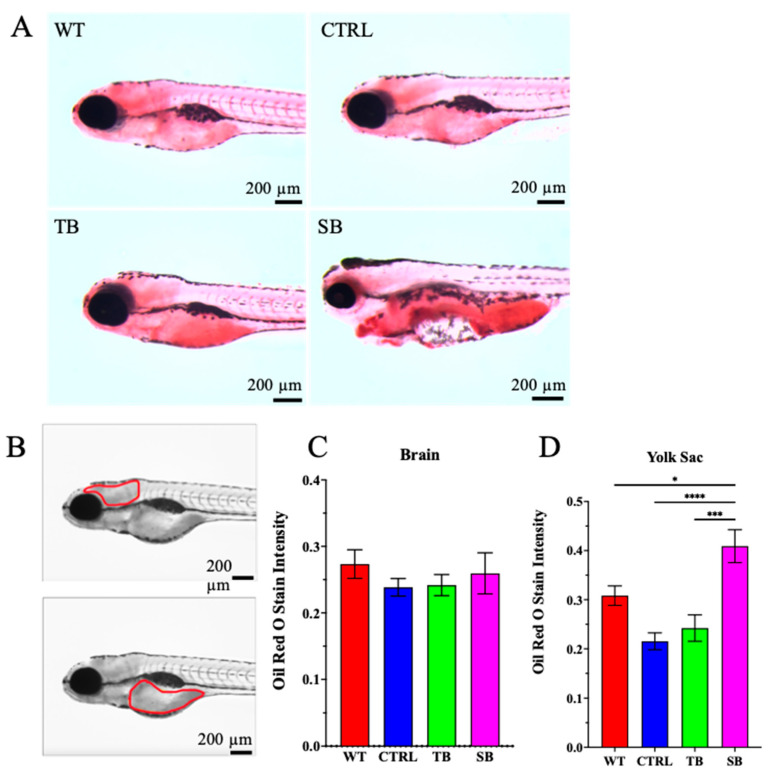
Lipid deposition in CPT2 knockdown larvae. (**A**). Oil Red O whole-mount staining was performed on 5 dpi larvae to evaluate lipid deposition. Scale bar = 200 µm. (**B**). Greyscale images were quantified for pixel intensity in brain and yolk sac to determine intensity of Oil Red O stain. Scale bar = 200 µm. (**C**–**D**). Oil Red O stain intensity in brain and yolk sac. (**A**–**D**). WT = wildtype; CTRL = control, scrambled MO-injected zebrafish; TB = translation blocking MO-injected zebrafish; SB = splice blocking MO-injected zebrafish. N ≤ 10 zebrafish per condition. * *p* ≤ 0.05, *** *p* < 0.001, **** *p* < 0.0001. Error bars = SEM.

**Figure 6 biomolecules-14-00914-f006:**
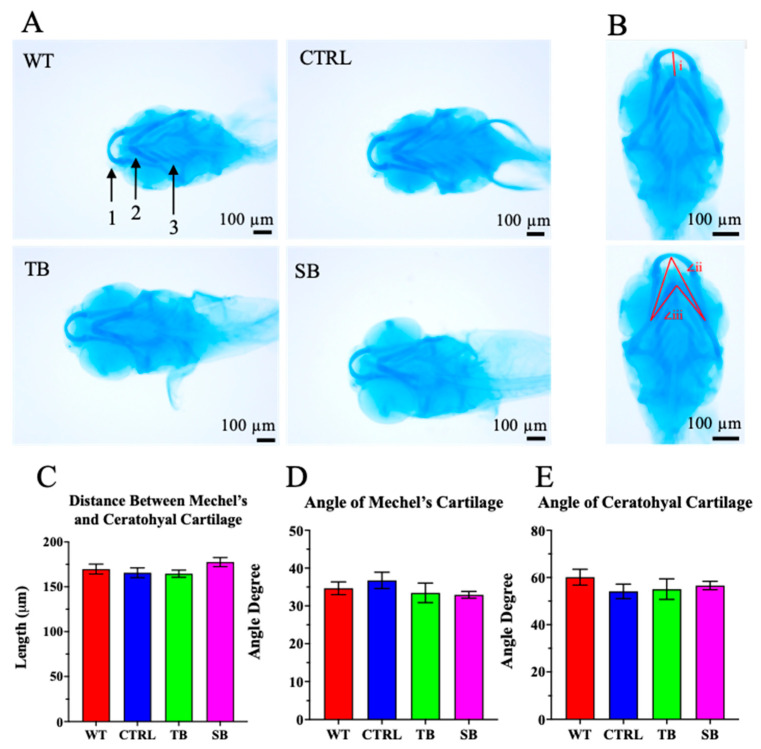
Cartilage development in CPT2 knockdown larvae. (**A**). Alcian blue stain was performed on 5 dpi larvae to evaluate cartilage development. Areas of focus were Mechel’s cartilage (1), ceratohyal cartilage (2), and ceratobranchial cartilage i–v (3). Scale bar = 100 µm. (**B**). Cartilage development was assessed on distance between Mechel’s and ceratohyal cartilage (i), angle of Mechel’s cartilage (∠ii), and angle of ceratohyal cartilage (∠iii). (**C**,**D**). Quantification of cartilage measurements. (**A**–**E**). WT = wildtype; CTRL = control, scrambled MO-injected fish; TB = translation blocking MO-injected zebrafish; SB = splice blocking MO. N ≤ 11 zebrafish per condition. Error bars = SEM.

**Figure 7 biomolecules-14-00914-f007:**
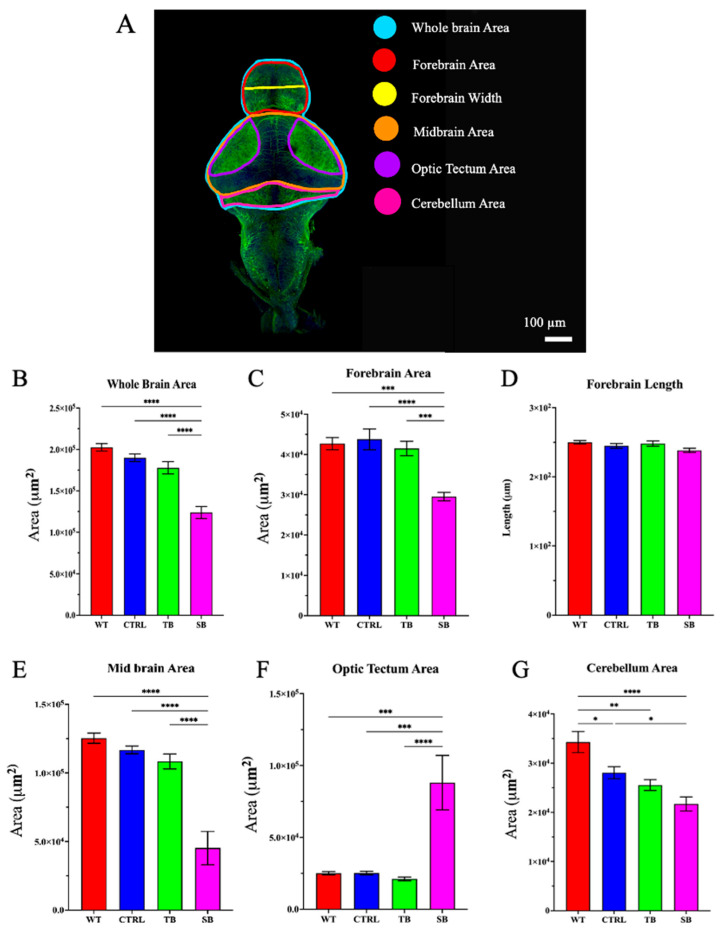
Brain development in CPT2 knockdown larvae. (**A**). Dissected zebrafish brain immunostained with acetylated tubulin. Schematic map of brain regions that were measured. Scale bar = 100 µm. (**B**–**G**). Quantification of area and length measurements from dissected and acetylated tubulin-stained brains. WT = wildtype zebrafish; CTRL = control, scrambled MO-injected zebrafish; TB = translation blocking MO-injected zebrafish; SB = splice blocking MO-injected zebrafish. N = 8–16 zebrafish brains per condition. * *p* ≤ 0.05, ** *p* < 0.01, *** *p* < 0.001, **** *p* < 0.0001. Error bars = SEM.

**Figure 8 biomolecules-14-00914-f008:**
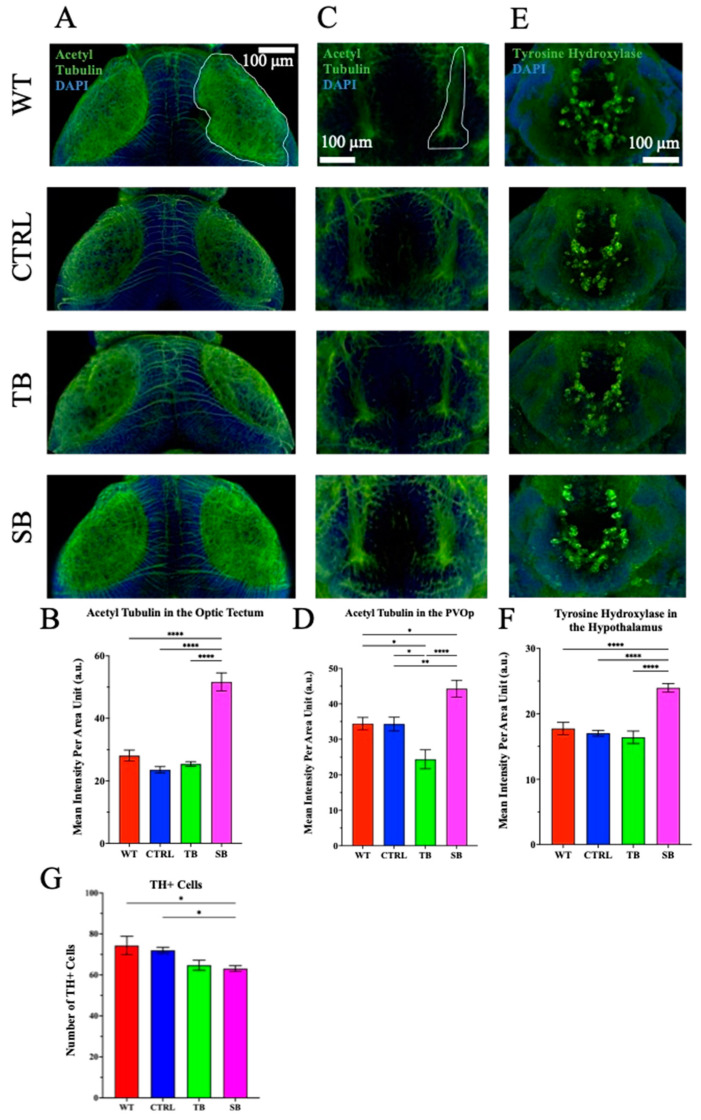
Immunofluorescence of neuronal network structure in CPT2 knockdown larvae brains. Brains were labeled with acetylated tubulin (green) and counterstained with DAPI. (**A**,**B**). The optic tectum is outlined in white and viewed on the dorsal side of the larvae. (**C**,**D**). The paraventricular organ posterior part (PVOp) region of the brain is outlined in white on the ventral side of the larvae. N = 8–16 per condition. (**E**,**F**). Brains were immunostained for tyrosine hydroxylase (green) and DAPI. The hypothalamus on the ventral side of the larva brain is shown. (**G**). Tyrosine hydroxylase-positive cells were counted from the hypothalamus. (**A**,**C**,**E**). Scale bar = 100 µm. (**A**–**G**). WT = wildtype, CTRL = control MO, TB = translation blocking MO, SB = splice blocking MO. N = 8–10 zebrafish brains per condition. * *p* ≤ 0.05, ** *p* < 0.01, **** *p* < 0.0001. Error bars = SEM.

**Figure 9 biomolecules-14-00914-f009:**
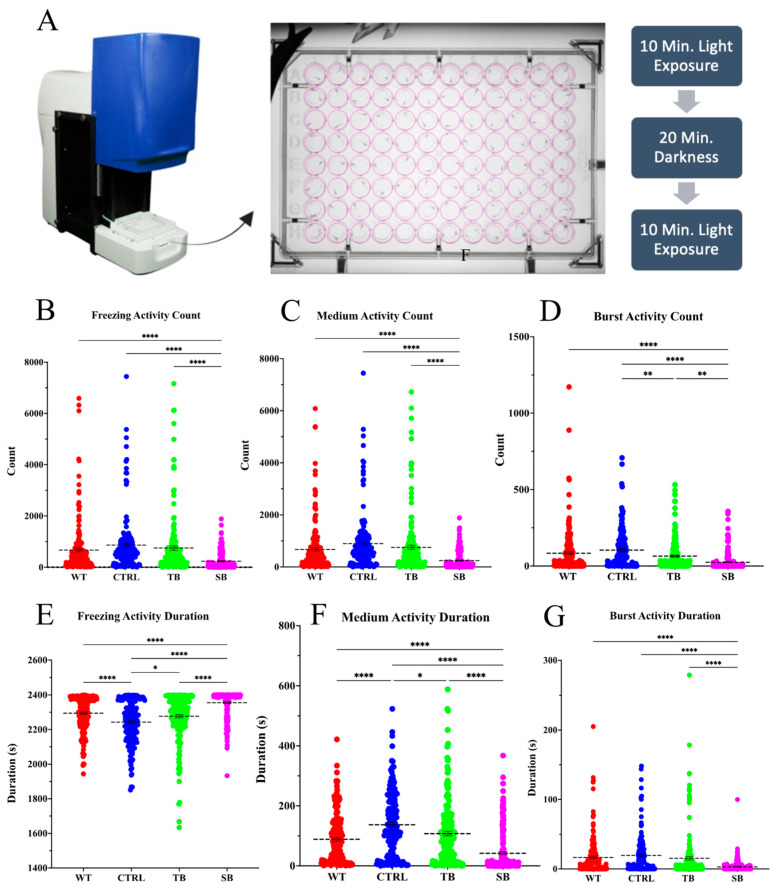
Swimming speeds in CPT2 knockdown larvae. (**A**). Behavior assays performed in ViewPoint Zebrabox with individual fish in 96-well plate for 40 min with light stimulation: 10 min light, 20 min dark, 10 min light. (**B**–**G**). Quantification of quantization assay measuring larvae freezing and swimming activity count and duration. WT = wildtype, CTRL = control MO, TB = translation blocking MO, SB = splice blocking MO. N = 96 fish used per condition for each of two trials. Dotted black lines = mean value. * *p* ≤ 0.05, ** *p* < 0.01, **** *p* < 0.0001. Error bars = SEM.

**Figure 10 biomolecules-14-00914-f010:**
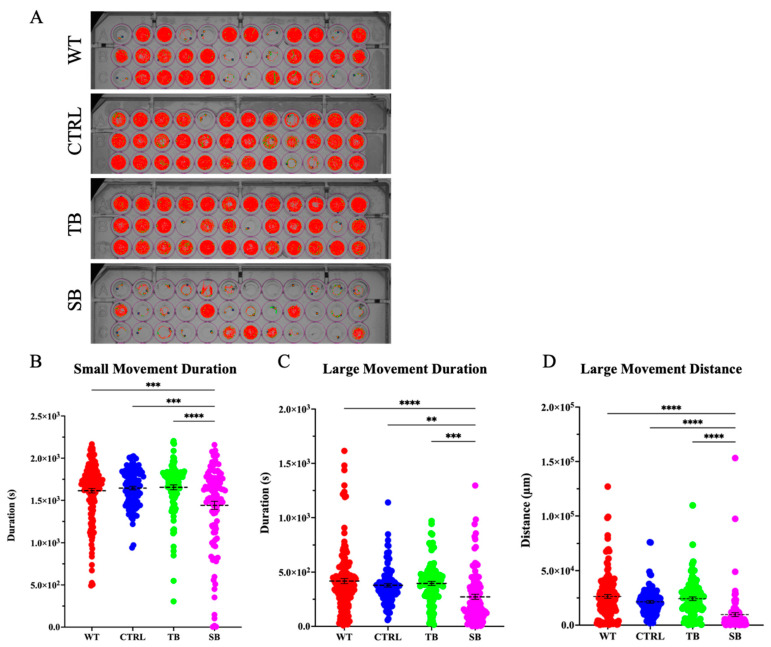
Swimming distance in CPT2 knockdown larvae. Tracking behavioral assay was performed using ViewPoint Zebrabox for 40 min with changing light stimulation; 10 min of light, 20 min of darkness, 10 min of light. (**A**). Tracked movement indicated by red (tracks small movement) and green (tracks large movement) lines for individual larvae during 40 min recording. (**B**–**D**). Quantification of tracking assay of larvae movement duration and distance. (**A**–**D**). WT = wildtype, CTRL = control MO, TB = translation blocking MO, SB = splice blocking MO. N = 60 fish used per condition for each of two trials. ** *p* < 0.01, *** *p* < 0.001, **** *p* < 0.0001. Dotted black lines = mean value. Error bars = SEM.

**Figure 11 biomolecules-14-00914-f011:**
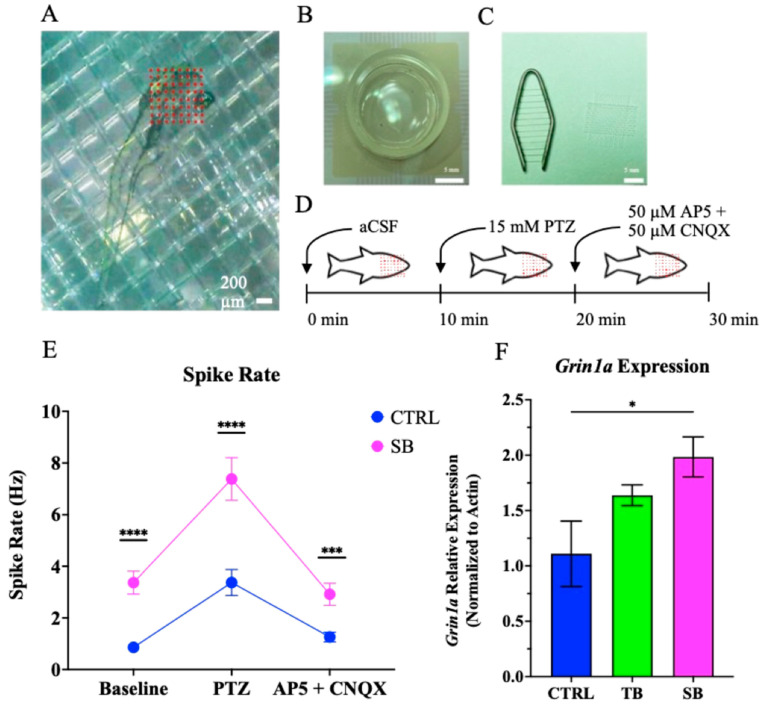
Seizure susceptibility and Grin1a expression. (**A**). Secured, living 5 dpi zebrafish on MED64 electrode. Scale bar = 200 µm. (**B**). MED64 electrode dish. Scale bar = 5 mm. (**C**). Tissue harp and mesh for securing larvae to electrode. Scale bar = 5 mm. (**D**). Schematic of 30 min electrophysiology recording, 10 min of artificial cerebrospinal fluid (aCSF) perfusion, 10 min of Pentylenetetrazol (PTZ) perfusion, and 10 min of 2(R)-amino-5-phosphonopentanoate and cyanquixaline (AP5 + CNQX) perfusion. (**E**). Spike rate (Hz) of CTRL and SB larvae. N = 3 zebrafish per condition. (**F**). *Grin1a* gene expression evaluated through RT-qPCR. N = 30 fish per condition for each of four trials. (**E**–**F**). CTRL = control MO, TB = translation blocking MO, SB = splice blocking MO. * *p* ≤ 0.05, *** *p* < 0.001, **** *p* < 0.0001. Error bars = SEM.

**Figure 12 biomolecules-14-00914-f012:**
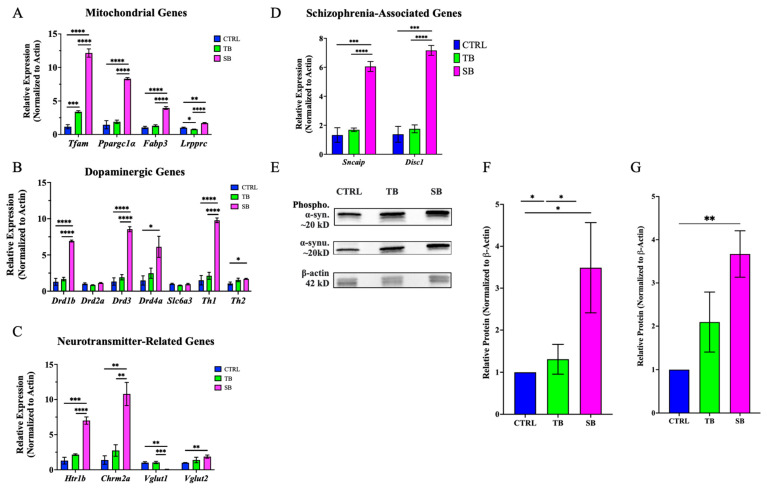
Gene expression in CPT2 knockdown larvae. (**A**). Mitochondrial gene expression. (**B**). Dopaminergic gene expression. (**C**). Neurotransmitter-related gene expression. (**D**). Schizophrenia-linked gene expression. (**E**). Western blot of phosphorylated and unphosphorylated α-synuclein compared to actin. (**F**). Quantification of unphosphorylated α-synuclein protein expression. (**G**). Quantification of phosphorylated α-synuclein protein expression. (**B**–**D**). WT = wildtype zebrafish; CTRL = control, scrambled MO-injected zebrafish; TB = translation blocking MO-injected zebrafish; SB = splice blocking MO-injected zebrafish. N = 30 zebrafish per condition for each of four trials. * *p* ≤ 0.05, ** *p* < 0.01, *** *p* < 0.001, **** *p* < 0.0001. Error bars = SEM.

## Data Availability

Data are available upon request to the corresponding author.

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
