# Peer review of "CPT2 Deficiency Modeled in Zebrafish: Abnormal Neural Development, Electrical Activity, Behavior, and Schizophrenia-Related Gene Expression"

_biomolecules, 2024, doi:10.3390/biom14080914_

Round 1

Reviewer 1 Report

Comments and Suggestions for Authors

To Authors:

In this paper entitled, “CPT2 Deficiency Modeled in Zebrafish: Abnormal Neural Development, Electrical Activity, Behavior, and Schizophrenia-Related Gene Expression”, by Baker et al, described about the zebrafish model of CPT2 deficiency generated by means of morpholino oligo-mediated gene knockdown of zebrafish cpt2 gene. They found that cpt2-deficient zebrafish larvae show abnormal body shape, reduced some brain structures, and enlarged optic tectum as morphological phenotypes, as well as increased several fatty-acylcarnitine species as a metabolic change. They also found that some behavioral activity was compromised in the cpt2-deficient larvae along with altered expression of serotonergic, cholinergic, glutamatergic genes and also schizophrenia-related genes. Their findings suggest possible underlying mechanisms of the symptoms in the patients so that help us to understand the basis of this disease. I have some comments and questions regarding the manuscript before publication.

1) The authors showed that the expression of Cpt2 protein is reduced in TB-MO and SB-MO injected larvae. I think Fig.2B is not convincing for TB.

2) In Fig.7, the authors showed that the size of some brain parts is altered and suggested possible cell death or reduced proliferation of neuronal cells. This part seems to be very important so adding some more direct evidence would be nice if possible.

3) The authors claimed that cpt2-deficient larvae (especially SB-MO injected ones) shows behavioral changes, however, the authors also showed that the larvae displayed abnormal body shape possibly by impaired muscle development. If muscle is underdeveloped, the behavioral change might not solely due to the neuronal dysfunction. Please explain.

4) In the introduction (line 75), the authors described as “The Cpt2 gene sequence in humans (ENSG00000157184) and zebrafish (ENSDARG00000038618) is 69% homologous”, but this description would be inappropriate. Must be “human and zebrafish Cpt2 proteins (or coding sequences) have 69% sequence similarity” or something like that.

5) Overall, I find this paper exceedingly difficult to read, because of many redundant descriptions and unnecessary, erroneous spaces and numbers are found in many places. The manuscript must be reshaped to be more readable to readers.

6) In many figures, asterisks showing p-values are missing.

Minor points:

line 33: One in 800 births sounds to be not so rare. I just wonder if this number is correct.

line 109-113: There are some redundant descriptions.

line 199-200: a-synuclein was written twice.

line 220: must be “product ions at m/z 85 (m/z should be italicized).

Line315: Missing parenthesis.

Line 333: Supposed to be “GABA”.

Line 580-584: irregular insertions of numbers “926” “927” “928” “929”.

Line 774-776: irregular number insertions.

Line 764: must be “upregulated”.

Line 803: supposed to be “transcripts” not “translation”.

As I explained above, there are unnecessary spaces are found in many places such as in line 57, 60, 69, 278. Please reshape the manuscript more carefully.

Author Response

Dear Reviewer #1. Thank you for your review and excellent comments. Your review of our manuscript led to significant improvements.

Our replies are found below.

1) The authors showed that the expression of Cpt2 protein is reduced in TB-MO and SB-MO injected larvae. I think Fig.2B is not convincing for TB.

Thank you, we agree the knockdown of the TB MO is mild, while SB MO is moderate.  Quantification of the CPT2 protein from three western blots demonstrates a % reduction.  We have clarified in the text that TB MO is a model for mild reduction in CPT2 protein and for assessment on larval development. 

2) In Fig.7, the authors showed that the size of some brain parts is altered and suggested possible cell death or reduced proliferation of neuronal cells. This part seems to be very important so adding some more direct evidence would be nice if possible.

Thank you for this important comment.  We agree.  This study serves as “proof of concept” for future work.  Further development of this model and use of CRISPCas9 to examine proband specific CPT2 mutations following our previous study will examine proliferation and cell death of neurons and glia.

3) The authors claimed that cpt2-deficient larvae (especially SB-MO injected ones) shows behavioral changes, however, the authors also showed that the larvae displayed abnormal body shape possibly by impaired muscle development. If muscle is underdeveloped, the behavioral change might not solely due to the neuronal dysfunction. Please explain.

Yes you are correct. We have added statements in the discussion that a lack of proper muscular, vascular, ligamentous, and/or bone development may also influence behavior.

4) In the introduction (line 75), the authors described as “The Cpt2 gene sequence in humans (ENSG00000157184) and zebrafish (ENSDARG00000038618) is 69% homologous”, but this description would be inappropriate. Must be “human and zebrafish Cpt2 proteins (or coding sequences) have 69% sequence similarity” or something like that.

Yes, thank you. We have corrected this statement. The sentences are as follows: The zebrafish (Danio rerio) represents an ideal vertebrate model system for studying human metabolic disease [19,20].  Approximately 70% of human genes have at least one zebrafish orthologue [21]. Human Cpt2 (ENSG00000157184) and zebrafish cpt2 (ENSDARG00000038618) are homologous (zfin.org) and the sequences show 70.9% alignment as determined by Expasy (expasy.org).

5) Overall, I find this paper exceedingly difficult to read, because of many redundant descriptions and unnecessary, erroneous spaces and numbers are found in many places. The manuscript must be reshaped to be more readable to readers.

We have edited the paper significantly to make the text concise and free of erroneous spaces and other errors.

6) In many figures, asterisks showing p-values are missing.

 Thank you we have corrected this throughout.

line 33: One in 800 births sounds to be not so rare. I just wonder if this number is correct.

This sentence has been corrected. One in 800 live births refers the cumulative occurrence of all inherited metabolic disorders.  Individually  inherited metabolic disorders are rare.  The myopathic forms of CPT2 are not considered rare in adults so the term rare has been deleted.

All other minor points, indicated by lines have been corrected. Thank you for your thorough comments.

line 109-113: There are some redundant descriptions.

line 199-200: a-synuclein was written twice.

line 220: must be “product ions at m/z 85 (m/z should be italicized).

 Line315: Missing parenthesis.

 Line 333: Supposed to be “GABA”.

 Line 580-584: irregular insertions of numbers “926” “927” “928” “929”.

 Line 774-776: irregular number insertions.

 Line 764: must be “upregulated”.

 Line 803: supposed to be “transcripts” not “translation”.

Reviewer 2 Report

Comments and Suggestions for Authors

The work describes a study in zebrafish using morpholino knockdown of ctp2 as a model for CPT2 deficiency in human. The CPT2 gene is associated with brain development and schizophrenia, which the others aim to study in zebrafish. The overall aim is interesting and worth studying. While schizophrenia itself is difficult (if not impossible) to study in zebrafish, neurodevelopmental traits and behavioural traits could emerge that aid in the understanding of the role of CPT2 in the human brain.

First of all, the authors use morpholino knockdown, which has gotten and negative image since the onset of CRISPR technology. To me, a properly conducted and controlled MO study can still be worth publishing. The authors provide a detailed description of their approach, and use two MOs for cross-validation of their findings. Unfortunately, the following parts make precisely those mistakes that give MO studies a bad name. 

In Figure 1, the authors show embryo viability post injection. values for wild-type are already low, but viability should not increase with higher MO dose. This indicates that technical aspects have a big influence in this study. Typical assays would have 80-90% viability across conditions with increase presence/severity of phenotypes with increasing doses. 

In Figure 2, the first control of successful MO activity (splice modulation) appears to fail. There is not evidence of splice modulation. The level of ctp2 mRNA seems to decrease, but this could be due to MO toxicity. If this is NMD (which would be in favour of the authors), this should be shown with an NMD-blocking strategy. 

The amount of reduction in Ctp2 protein levels is low in the TB MO group, and reasonable in the SB group. all in all, the paper should provide more even of successful knockdown before phenotypic investigations are warranted. 

Ideally showing a clear effect on transcript level (exon skipping) combined with phenotypic rescue studies by mRNA injection for both MOs. Without that, all phenotypic could be off-target phenotypes.

In figure 3 and 4, the morphological phenotypes is highly reminiscent of MO toxicity. Showing successful mRNA rescue for these phenotypes would already be highly convincing the phenotype is real. In addition, I would like to see rescue for key schizophrenia phenotypes. Not mandatory, but adding to the proof that the results are on-target, rescue by metabolic supplementation could be considered. 

Alternatively, a knock-out line could be used to avoid the risks associated with MOs. 

With uncertainty of the value of the phenotypes, it is of little use to discuss the interpretation. Some might not be rescued, and therefore not the result of loss of Ctp2 function. 

There are some interesting result in the extensive phenotyping results (if they appear true) that do make that the work has the potential for a good publication: behavioural changes, changes in mitochondrial and neurotransmitter gene expression, brain anatomy and acyl-carnitine levels. 

I am surprised to see schizophrenia associated genes upregulation in the zebrafish models. Normally, genes are associated with disease based on loss-of-function. The authors should address this in the discussion. 

line 75: Ctp2 is not written in human (CTP2) or zebrafish (cpt2) nomenclature. This mistake is made in the entire manuscript. For details, see https://zfin.atlassian.net/wiki/spaces/general/pages/1818394635/ZFIN+Zebrafish+Nomenclature+Conventions

line 128: "integration" should be "uptake". MOs should not integrate in the genome (which is wat the words suggest))

Comments on the Quality of English Language

/

Author Response

Reviewer 2  Thank you for your comments and suggestions.  We have responded to your review with comments below.  The comments that we are able to address have significantly improved the manuscript. 

First of all, the authors use morpholino knockdown, which has gotten and negative image since the onset of CRISPR technology. To me, a properly conducted and controlled MO study can still be worth publishing. The authors provide a detailed description of their approach, and use two MOs for cross-validation of their findings. Unfortunately, the following parts make precisely those mistakes that give MO studies a bad name. 

CRISPR technology is certainly useful and future studies are aimed and using CRISPER to model the proband’s specific mutations as described in our previous study (https://doi.org/10.3390/reports3040031) using the zebrafish model system.  We plan to use the high throughput capacity of the zebrafish to examine the effect of the patient specific mutation and investigate the capaticy to rescue the knockout.   Since CRISPER does not allow for "tuning down" of protein expression, this study examines MO knockdown as a proof of concept study investigating mild-to-moderate cpt2 deficiency. We revised the introduction to explicitly state the rationale for starting these studies using the MO strategy.

Morpholino oligonucleotides (manufactured by GeneTools) are the most widely used antisense knockdown tool in zebrafish embryos (Heasman, 2002; Nasevicius & Ekker, 2000) and  Peter M. Eimon, in Methods in Enzymology, 2014

In Figure 1, the authors show embryo viability post injection. values for wild-type are already low, but viability should not increase with higher MO dose. This indicates that technical aspects have a big influence in this study. Typical assays would have 80-90% viability across conditions with increase presence/severity of phenotypes with increasing doses. 

TuAB wild type fish exhibit a range of baseline viability depending on clutch size and housing.  Death of fertilized eggs is documented to range from 20-30% depending on housing, water quality, variability (doi: 10.1089/zeb.2011.0688; doi: 10.3390/ijms222413417).   Fertilized eggs for our studies are placed in cell culture dishes for experimentation and WT viability is at 77%. We conclude that our WT variability is within the normal range.  The highest concentration of MO examined is 1.0mM and is not more viable than that of 0.5mM.  Reduced viability at 0.25mM for CTRL MOs was removed to focus our presentation of data as recommended by another reviewer.

In Figure 2, the first control of successful MO activity (splice modulation) appears to fail. There is not evidence of splice modulation. The level of ctp2 mRNA seems to decrease, but this could be due to MO toxicity. If this is NMD (which would be in favour of the authors), this should be shown with an NMD-blocking strategy. Ideally showing a clear effect on transcript level (exon skipping) combined with phenotypic rescue studies by mRNA injection for both MOs. Without that, all phenotypic could be off-target phenotypes.

The activity of splice-blocking cpt2 morpholinos can be detected by RT-PCR.  Successful splice modification will result in a mobility shift or complete loss of the wild-type transcript as published by Draper, Morcos, & Kimmel, 2001.   Splice-blocking morpholinos should only target zygotic transcripts and we observed through RTPCR that near complete loss of the wild-type transcript was occurred.  This result is supported by a significant decrease in CPT2. In contrast, translation-blocking morpholinos can inhibit both zygotic and maternal mRNA transcripts.  NMD-blocking strategies would be additional confirmatory studies. We indicate that these experiences can be performed in future studies.

The amount of reduction in Ctp2 protein levels is low in the TB MO group, and reasonable in the SB group. all in all, the paper should provide more even of successful knockdown before phenotypic investigations are warranted. 

Thank you for your comment. We have clarified throughout that we were aiming to develop a model system to investigate mild to moderate Cpt2 deficiency and the low reduction in (TB) and reasonable reduction in SB of CPT2 protein allows us to investigate a reduction in CPT2 protein that is more analogous to a mild to moderate loss of CPT2 function.  A total loss of CPT2 would likely be lethal, similar to that seen in humans.

In figure 3 and 4, the morphological phenotypes is highly reminiscent of MO toxicity. Showing successful mRNA rescue for these phenotypes would already be highly convincing the phenotype is real. In addition, I would like to see rescue for key schizophrenia phenotypes. Not mandatory, but adding to the proof that the results are on-target, rescue by metabolic supplementation could be considered. Alternatively, a knock-out line could be used to avoid the risks associated with MOs. 

Thank you for the excellent suggestions. We have included in the discussion of the data that following this characterization of the model for mild to moderate CPT2 deficiency, we intend to focus future studies on investigating patient specific mutations, specifically the proband with diagnosed schizophrenia, and develop CRISPR patient specific mutation. Rescue studies will be implemented in this model system and compared to our TB and SB models.

With uncertainty of the value of the phenotypes, it is of little use to discuss the interpretation. Some might not be rescued, and therefore not the result of loss of Ctp2 function. There are some interesting result in the extensive phenotyping results (if they appear true) that do make that the work has the potential for a good publication: behavioural changes, changes in mitochondrial and neurotransmitter gene expression, brain anatomy and acyl-carnitine levels. I am surprised to see schizophrenia associated genes upregulation in the zebrafish models. Normally, genes are associated with disease based on loss-of-function. The authors should address this in the discussion. 

Yes we agree some phenotypes may not be rescued.  Future experiments will be rescue experiments as you suggest.  As well, an Aim of our grant is to use CRISPR constructs to model the proband specific mutation for CPT2 in zebrafish.  MOs were a logical first step and one with “tittering” capabilities.  Both clarifications are in the discussion.

While zebrafish will not mimic humans, they are a vertebrate system with high throughput capabilities, clearly defined behaviors, known gene expression and have been shown to be a useful model for mechanistic and pharmacokinetic studies for schizophrenia. Examples:

  1. Developing zebrafish experimental animal models relevant to schizophrenia. Konstantin A Demin , Darya A Meshalkina , Andrey D Volgin , Oleg V Yakovlev, Murilo S de Abreu, Polina A Alekseeva, Ashton J Friend, Anton M Lakstygal, Konstantin Zabegalov, Tamara G Amstislavskaya, Tatyana Strekalova , Wandong Bao, Allan V Kalueff, Neurosci Biobehav Rev 2019 Oct:105:126-133. doi: 10.1016/j.neubiorev.2019.07.017. Epub 2019 Jul 29.
  2. PMID: 31369798 DOI: 10.1016/j.neubiorev.2019.07.01
  3. The Role of Zebrafish and Laboratory Rodents in Schizophrenia Research. Langova V, Vales K, Horka P, Horacek J.Front Psychiatry. 2020 Aug 27;11:703. doi: 10.3389/fpsyt.2020.00703. eCollection 2020.PMID: 33101067
  4. Zebrafish as a tool to study schizophrenia-associated copy number variants. Campbell PD, Granato M.Dis Model Mech. 2020 Apr 29;13(4):dmm043877. doi: 10.1242/dmm.043877.PMID: 32433025
  5. Glutamate NMDA Receptor Antagonists with Relevance to Schizophrenia: A Review of Zebrafish Behavioral Studies. Benvenutti R, Gallas-Lopes M, Marcon M, Reschke CR, Herrmann AP, Piato A.Curr Neuropharmacol. 2022 Mar 4;20(3):494-509. doi: 10.2174/1570159X19666210215121428.PMID: 3358873

There are many genes that are upregulated in human schizophrenic tissue samples and in model systems.  Searching pubmed for schizophrenia, gene expression, and upregulation results in over 650 references. Some of these studies identify increased expression of mutant forms of protein in schizophrenia. Other studies use Meta- analyses or other Big Data analyses to evaluate schizophrenia and related synucleinopathies and demonstrate a host of up and downregulated genes associated with specific disease states.  Examples:

    • “Deep proteomics identifies shared molecular pathway alterations in synapses of patients with schizophrenia and bipolar disorder and mouse model.” Aryal S, Bonanno K, Song B, Mani DR, Keshishian H, Carr SA, Sheng M, Dejanovic B.Cell Rep. 2023 May 30;42(5):112497. doi: 10.1016/j.celrep.2023.112497. Epub 2023 May 11.PMID: 37171958
  • Activity-Dependent Changes in Gene Expression in Schizophrenia Human-Induced Pluripotent Stem Cell Neurons.Roussos P, Guennewig B, Kaczorowski DC, Barry G, Brennand KJ.JAMA Psychiatry. 2016 Nov 1;73(11):1180-1188. doi: 10.1001/jamapsychiatry.2016.2575.PMID: 27732689 Free PMC article.
  • “DISC1-Δ3, a major DISC1 variant that lacks exon 3,” PMID: 36131044;
  • “Phenotypic Landscape of Schizophrenia-Associated Genes Defines Candidates and Their Shared Functions” Cell. 2019 Apr 4;177(2):478-491.e20. doi: 10.1016/j.cell.2019.01.048. Epub 2019 Mar 28.Summer B Thyme, Lindsey M Pieper , Eric H Li , Shristi Pandey , Yiqun Wang , Nathan S Morris , Carrie Sha , Joo Won Choi , Kristian J Herrera , Edward R Soucy , Steve Zimmerman , Owen Randlett Joel Greenwood, Steven A McCarroll , Alexander F Schier  PMID: 30929901 PMCID: PMC6494450 DOI: 10.1016/j.cell.2019.01.048

Other comments: line 75: Ctp2 is not written in human (CTP2) or zebrafish (cpt2) nomenclature. This mistake is made in the entire manuscript. For details, see https://zfin.atlassian.net/wiki/spaces/general/pages/1818394635/ZFIN+Zebrafish+Nomenclature+Conventions

Thank you. We have corrected all zebrafish references to cpt2 nomenclature when referring to the zebrafish gene, Cpt2 when referring to the huma gene and CPT2 when referring to protein.

line 128: "integration" should be "uptake". MOs should not integrate in the genome (which is wat the words suggest))

Yes, corrected. Thank you.

Reviewer 3 Report

Comments and Suggestions for Authors

A well executed, highly complex, but well designed integrated pathophysiological study on human Schizophrenia disorder, using the zebrafish model.

Please address comments in the discussion section and fix up all the gaps in the formatting of the text.

Comments on the Quality of English Language

Author Response

Thank you for your editing. You work has significantly improved the manuscript.

We have updated the manuscript and addressed your comments. Additional references and a paragraph on environmental toxins have been added to the discussion as recommended.